# A Focus on the Pathophysiology of Adrenomedullin Expression: Endothelitis and Organ Damage in Severe Viral and Bacterial Infections

**DOI:** 10.3390/cells13110892

**Published:** 2024-05-22

**Authors:** Silvia Spoto, Stefania Basili, Roberto Cangemi, José Ramón Yuste, Felipe Lucena, Giulio Francesco Romiti, Valeria Raparelli, Josepmaria Argemi, Giorgio D’Avanzo, Luciana Locorriere, Francesco Masini, Rodolfo Calarco, Giulia Testorio, Serenella Spiezia, Massimo Ciccozzi, Silvia Angeletti

**Affiliations:** 1Diagnostic and Therapeutic Medicine Department, Fondazione Policlinico Universitario Campus Bio-Medico, Via Alvaro del Portillo, 200, 00128 Rome, Italy; giorgio.davanzo@policlinicocampus.it (G.D.); l.locorriere@policlinicocampus.it (L.L.); f.masini@policlinicocampus.it (F.M.); r.calarco@policlinicocampus.it (R.C.); g.testorio@policlinicocampus.it (G.T.); s.spiezia@policlinicocampus.it (S.S.); 2Department of Translational and Precision Medicine, Sapienza University, Viale dell’Università, 30, 00185 Rome, Italy; stefania.basili@uniroma1.it (S.B.); roberto.cangemi@uniroma1.it (R.C.); valeria.raparelli@uniroma1.it (V.R.); 3Division of Infectious Diseases, Faculty of Medicine, Clinica Universidad de Navarra, University of Navarra, Avda. Pío XII, 36, 31008 Pamplona, Spain; jryuste@unav.es; 4Department of Internal Medicine, Faculty of Medicine, Clinica Universidad de Navarra, University of Navarra, Avda. Pío XII, 36, 31008 Pamplona, Spain; 5Departamento de Medicina Interna, Clinica Universidad de Navarra, Avda. Pío XII, 36, 31008 Pamplona, Spain; flucena@unav.es (F.L.); jargemi@unav.es (J.A.); 6Unit of Medical Statistics and Molecular Epidemiology, Università Campus Bio-Medico di Roma, 00128 Rome, Italy; m.ciccozzi@unicampus.it; 7Unit of Laboratory, Fondazione Policlinico Universitario Campus Bio-Medico, Via Alvaro del Portillo, 200, 00128 Rome, Italy; s.angeletti@policlinicocampus.it; 8Research Unit of Clinical Laboratory Science, Department of Medicine and Surgery, Università Campus Bio-Medico di Roma, Via Alvaro del Portillo, 21, 00128 Rome, Italy

**Keywords:** adrenomedullin (ADM), mid-regional proadrenomedullin (MR-proADM), endothelitis, sepsis, viral infection, biomarkers

## Abstract

Adrenomedullin (ADM) is a peptide hormone produced primarily in the adrenal glands, playing a crucial role in various physiological processes. As well as improving vascular integrity and decreasing vascular permeability, ADM acts as a vasodilator, positive inotrope, diuretic, natriuretic and bronchodilator, antagonizing angiotensin II by inhibiting aldosterone secretion. ADM also has antihypertrophic, anti-apoptotic, antifibrotic, antioxidant, angiogenic and immunoregulatory effects and antimicrobial properties. ADM expression is upregulated by hypoxia, inflammation-inducing cytokines, viral or bacterial substances, strength of shear stress, and leakage of blood vessels. These pathological conditions are established during systemic inflammation that can result from infections, surgery, trauma/accidents or burns. The ability to rapidly identify infections and the prognostic, predictive power makes it a valuable tool in severe viral and bacterial infections burdened by high incidence and mortality. This review sheds light on the pathophysiological processes that in severe viral or bacterial infections cause endothelitis up to the development of organ damage, the resulting increase in ADM levels dosed through its more stable peptide mid-regional proadrenomedullin (MR-proADM), the most significant studies that attest to its diagnostic and prognostic accuracy in highlighting the severity of viral or bacterial infections and appropriate therapeutic insights.

## 1. Introduction

Adrenomedullin (ADM) is a hormone isolated in 1993 from pheochromocytoma cells [1] but synthesized ubiquitously in the body, whose main function is vasodilation [1,2] and preservation of endothelial barrier integrity [3,4].

ADM also plays cardiokinetic [5], bronchodilator, diuretic [6], antioxidant [5], angiogenic [7], antiapoptotic [6,8], antifibrotic [7,9], immunoregulatory [10], and antimicrobial [11] roles. ADM expression is upregulated by hypoxia, inflammation-inducing cytokines, viral or bacterial substances, shear stress strength [12], and blood vessel leakage [5,13,14,15,16,17]. These pathological conditions set in during systemic inflammation that may result from infection, surgery [18], trauma/accident, or burns.

The ADM value expresses the endothelial damage that correlates with the severity of organ damage that occurs in severe infections [19,20,21,22] and particularly in the most severe clinical manifestations of infections, such as acute respiratory distress syndrome (ARDS) and multiorgan dysfunction syndrome (MODS) [23]. Related to organ damage, it also correlates with major clinical severity scores, such as the Systemic Inflammatory Response Syndrome (SIRS) criteria, the quick Sequential Organ Failure Score (q-SOFA), the Sequential Organ Failure Assessment (SOFA), the National Early Warning Score (NEWS), the Modified Early Warning Score (MEWS), or the Acute Physiology and Chronic Health Evaluation II (APACHE II) [20,24,25].

This biomarker increases rapidly in severe viral or bacterial infections, indicating their severity, prognosis, proper and timely management, and the most appropriate care setting [7,26,27,28]. Serum levels of ADM tend to be higher in patients with severe bacterial infections than in those with viral infections [29,30,31,32,33], guiding and limiting antibiotic prescribing according to antimicrobial stewardship directives to reduce antibiotic resistance and improve patient outcomes [34]. Therefore, given the high level of diagnostic, prognostic, early-elevation sensitivity, proportionate cost, rapid turnaround time [35] of this biomarker, and the incidence and mortality of serious infections such as Influenza [36,37] and SARS-CoV-2 [38] or sepsis [39], it represents a significant clinical aid.

Indeed, considering that out of about one billion cases of seasonal Influenza per year, at least three to five million are severe, causing 290,000 to 650,000 respiratory deaths each year [36,37] and that the more than 772 million confirmed cases of coronavirus disease (COVID-19) caused more than 6.9 million deaths globally [38] and that ADM has been shown to be the best biomarker for the stratification of mortality risk in critically ill patients with COVID-19 [28], its essential value is appreciated. 

On the other hand, sepsis is the clinical expression of a severe bacterial infection characterized by an exaggerated response of the immune system to the presence of bacteria, resulting in widespread inflammation and tissue damage [39], the incidence of which is progressively increasing. Indeed, in the past 10 years, the incidence rate of sepsis has been 437 cases of sepsis and 270 cases of severe sepsis per 100,000 person-years with hospital mortality of 17% or 26% for sepsis or severe sepsis, respectively, with estimated costs of more than $24 billion per year [39]. In this context, ADM has shown optimal diagnostic and prognostic accuracy of sepsis, allowing early diagnosis, expressing severity or poor outcome, stratifying septic patients [40], indicating the most appropriate management, including with potential monoclonal antibodies treatments aimed at blocking endothelial damage responsible for microvascular damage, organ damage and potential death [23]. However, further research and validation of the role of ADM in infection management is needed to fully understand its potential and optimize its use.

This review sheds light on the pathophysiological processes that in severe viral or bacterial infections cause endothelitis up to the development of organ damage, the resulting increase in ADM levels dosed through its more stable peptide mid-regional proadrenomedullin (MR-proADM), the most significant studies that attest to its diagnostic and prognostic accuracy in highlighting the severity of viral or bacterial infections and appropriate therapeutic insights.

## 2. Physiology of Adrenomedullin

ADM is a 6-kilodalton (52 amino acid) peptide hormone isolated in 1993 from a pheochromocytoma whose gene is located at a single locus on chromosome 11 and consists of four exons and three introns [41,42,43,44].

ADM, as well as calcitonin, procalcitonin (PCT) and amylin, belong to the calcitonin gene superfamily of peptides (CGRP) [45,46,47].

Its messenger RiboNucleic Acid (mRNA) encodes information for the synthesis of a preprohormone known as preproadrenomedullin, of 185 amino acids, which is subsequently degraded to a 164-amino acid peptide called proadrenomedullin. The latter consists of three vasoactive peptides, ADM, proadrenomedullin aminoterminal peptide (PAMP) and adrenotensin, and a region with no known activity, MR-proADM [48].

It is synthesized by endothelial cells (ECs) and VSMCs in response to an excess of fluid volume in order to preserve the integrity of the endothelial barrier. It easily moves across the blood and interstitial spaces and attaches to various receptors found primarily in cardiovascular and pulmonary tissues [11,49,50]. It is important to emphasize that ADM mRNA expression in EC cultures reaches concentrations 20–40 times higher than in the adrenal gland where it was isolated [43,44].

What are its main functions? The main role of ADM is to promote vasodilation in both resistance and capacitance blood vessels, resulting in increased blood flow [5,6]. It further reduces vasoconstriction by inhibiting the renin-angiotensin-aldosterone system (RAAS) and helps maintain endothelial integrity by decreasing vascular permeability [5,6].

From the point of view of half-life, ADM has a short half-life of 22 min because it is rapidly eliminated from the bloodstream through two distinct processes: through the ADM-receptor complex that undergoes internalization and degradation and through protein-lytic degradation (endopeptidase) that eliminates it by degrading it [15,44,51,52]. ADM is degraded in the kidney, adrenal, and lung, whilst metalloproteinase inhibitors inhibit this degradation [53]. 

## 3. Adrenomedullin and Mid-Regional Pro-Adrenomedullin

Regarding the assay of blood concentration of ADM, which has a short half-life, it is more functional and practical to test the peptide obtained from ADM, with a longer half-life, the mid-regional pro-adrenomedullin (MR-proADM). The latter is a peptide composed of 48 amino acids, derived from the medial region of proADM and released in an equivalent proportion of 1:1 with ADM. MR-proADM provides a proportional indication of ADM levels and activity but has a longer half-life than ADM [35], remaining stable for at least 75 days in the absence of clinical changes [35].

MR-proADM analysis is determined by the automated B.R.A.H.M.S. KRYPTOR compact PLUS method (Thermo Fisher Scientific, Hennigsdorf, Germany) using the Time-Resolved Amplified Cryptate Emission (TRACE) technique. The detection limit of the assay is 0.05 nmol/L, and intra- and inter-assay coefficients of variation were under 4 and 11%, respectively [28,35]. The turnaround time is about an hour; the cost varies and is about USD 20/test. In healthy individuals, MR-proADM levels are around 0.33 nmol/L [35]. However, MR-proADM levels of 0.8 nmol/L are indicative of viral or bacterial infections, with higher values indicating more severe infections (1.2 to 1.9 nmol/L for bacterial localized infections and 3.7 nmol/L for sepsis or septic shock). In patients with sepsis and septic shock, MR-proADM levels above 3.4 and 4.3 nmol/L, respectively, are significantly associated with a higher risk of mortality within 90 days [35,54,55,56].

## 4. Adrenomedullin Physiologic Functions

Where is ADM synthesized? ADM is widely synthesized throughout the body, but it is particularly abundant in certain organs where it is expressed according to a descending gradient, such as the adrenal medulla, cardiac atria, lung, kidney, blood vessels, bone, adipose tissue, anterior pituitary, thalamus and hypothalamus [57,58]. ADM is synthesized by different cell types, comprising ECs, VSMCs, macrophages, and monocytes, after being exposed to inflammatory triggers like interleukin-1 (Il-1), tumor necrosis factor (TNF), or lipopolysaccharide (LPS) [1,27,43,59,60]. On the other hand, its receptors have been identified in blood vessels, heart, lungs, skeletal muscle, nerve tissue, and various cell types such as ECs, VSMCs, cardiomyocytes, macrophages, and dendritic cells [26,59,61,62,63,64,65]. Indeed, the receptors and binding sites for ADM were mostly represented within the cardiovascular and lung tissue [66]. ADM has specific AM and AM1 receptors that are deputed to angiogenesis and vascular homeostasis, while AM2 receptors are deputed to lymphatic vessel function [66]. Afterward, ADM attaches to its receptor, composed of a combination of calcitonin receptor-like receptor (CRLR) and receptor activity-modifying proteins 2 and 3 (RAMP2, 3) [1,67,68].

Below, we examine in detail the roles ADM plays. Its main role is as a vasodilator (Figure 1). Physiologically, as ADM is present both within ECs (intravascular) and in its interstitium and VSMCs, it plays this vasodilator role both intravascularly and interstitially. In vitro, at the intravascular level, it acts on ECs by improving vascular integrity and reducing vascular permeability [5]. At the intravascular level, ADM causes vessel release by stimulating the production of endothelial nitric oxide synthase (eNOS), resulting in vasodilation of VSMCs [1,68,69,70]. In addition, ADM acts directly on VSMCs by activating protein kinase A (PKA) through the cAMP pathway, resulting in the relaxation of VSMCs [1,68,69,70]. It also maintains vascular integrity by acting on endothelial cell junctions, decreasing actomyosin contraction and vascular permeability during severe inflammation [4,6,10,71].

At the interstitial level, on the other hand, ADM results in vasodilation through (a) inhibition of VSMC contraction, acting both directly on VSMCs and indirectly on ECs and (b) nitric oxide (NO) production [10,72]. Moreover, ADM inhibits the RAAS (antagonizing angiotensin II inhibiting aldosterone secretion) and oxidative stress and suppresses excessive tissue proliferation and secretion of catecholamines, adrenocorticotropic hormone (ACTH) and insulin [6,73]. Furthermore, ADM is a multipotent regulatory peptide with a number of biological activities, acting at the systemic level as a vasodilator, positive inotropic, diuretic, natriuretic, and bronchodilator [13].

Chronically, ADM also has antihypertrophic, anti-apoptotic, antifibrotic, antioxidant, and angiogenic effects [8,13,74]. Many studies showed that ADM has cardioprotective effects, reducing cardiac hypertrophy and fibrosis -modulating fibroblast proliferation and cardiac remodeling, and heart myocyte apoptosis, decreasing lung capillary wedge and arterial pressure, attenuating oxidative stress and increasing cardiac index, diuresis and natriuresis [75,76,77,78,79,80,81,82].

It stimulates the angiogenetic and lymphangiogenesis effect by reducing edema through Connexin 43 (Cx43) and is an inhibitor of tissue congestion [4,7,73,83,84,85,86,87,88,89,90,91,92,93,94,95].

Furthermore, ADM plays a decisive role in the development of endothelial barrier function [1,2,83,84]. How does ADM contribute to this crucial function? This occurs by actively inhibiting the formation of stress fibers, which exert tension on intercellular junctions through the cyclic adenosine monophosphate (cAMP)-PKA pathway, thus triggering Rap1 activation and suppressing RhoA/ROCK activity [96,97]. In this context, ADM functions by activating its Gs-coupled receptor, which is noteworthy as it is also affected by multiple agents that restrict endothelial permeability, including prostacyclin (PGI2), prostaglandin E2 (PGE2), and β-adrenergic agonists [1,98,99]. Receptor activations trigger an increase in intracellular levels of cAMP, which is facilitated by adenyl cyclase converting adenosine triphosphate (ATP) into cAMP [100,101]. Increased cAMP levels stimulate the activation of Epac1, which acts as a guanine nucleotide exchange factor for Rap1. This subsequently enhances the endothelial barrier function through various mechanisms [1,102,103]. In this context, the binding between Rap1 effector Rasip1 and the transmembrane receptor heart of glass (HEG1) is necessary [104]. Furthermore, Rap1 triggers the activation of Cdc42 and its effector MRCK through the recruitment of FDG5 at intercellular junctions, resulting in enhanced bundling of cortical actin and subsequently reinforcing the endothelial barrier [1,105].

A further remarkable feature is ADM immunoregulatory effect, reducing transcription of proinflammatory genes, increasing transcription of anti-inflammatory cytokines, inhibiting cytokine-induced neutrophil chemoattractant (CINC/CXCL-1), interleukin-1β (IL-1β) induced TNF-α secretion, Interleukin-6 (IL-6), macrophages and rat Kupffer cells [10,92,93].

Interestingly, ADM has antimicrobial properties by inhibiting bacterial and fungal growth [11]. Research has shown that both ADM and PAMP fragments exert such antimicrobial and antimycotic activity by opening hydrophilic channels in the bacterial-fungal membrane [11,48,49], which, by altering its permeability demonstrates potent microbial actions against both Gram-positive and Gram-negative bacteria and *Candida albicans* [11,49,50,51,58,94]. Both ADM and PAMP were found distributed on the surface of the colonic mucosa, showing dose-dependent antimicrobial activity against *E. coli* and thus suggesting the role of this peptide in mucosal defense [106]. More studies are needed to release the relationship between ADM and complement and adrenomedullin-binding protein-1 [107].

In line with these notions, it should be mentioned that from a physiological, pathophysiological, and therapeutic point of view, the distribution and concentration of ADM in the bloodstream, thus at the intravascular and interstitial levels, is of paramount importance. Under pathological conditions that cause endothelial damage, as in the case of severe viral or bacterial infections, there is an increase in the passage of ADM from the endovascular space to the interstitial space. This shift seems to create an imbalance in ADM’s protective versus detrimental effects. Indeed, ADM present in the circulation protects vascular barrier function, whereas that present in the interstitium leads to vasodilatation and impairs vascular barrier function [1]. Furthermore, the use of ADM infusion has been shown to have a detrimental therapeutic effect because it accentuates harmful vasodilatory effects in the already hypotensive septic shock patient. Moreover, ADM tends to adhere to artificial surfaces, thereby limiting its clinical use. Conversely, the infusion of anti-ADM antibodies, which block ADM at the interstitial level, is showing great promise [1,52].

## 5. Pathophysiology of Endothelitis: Adrenomedullin Expression

Why and how is ADM produced and expressed? ADM expression is upregulated by hypoxia, cytokines that induce inflammation, viral or bacterial substances, the force of shear stress [12], and the leakage of blood vessels are all factors that trigger the upregulation of ADM, which in turn contributes to the breakdown of ADM regulation [13,14,15,16,17]. All these pathological conditions set in during systemic inflammation that may result from infection, surgery, trauma/accidents, or burns [1].

TNF-α/β, IL-1-α/β and LPS stimulate ADM production and release, whereas steroid or thyroid hormones, angiotensin II, norepinephrine, substance P, endothelin-1 and bradykinin stimulate its production and secretion [108]. On the other hand, transforming growth factor (TGF)-β, interferon γ, thrombin, forskolin, and 8-bromo-cAMP inhibit ADM synthesis [109].

It is essential to note that systemic inflammation causes endothelial damage [110,111] by acting on the single layer of endothelial cells that line blood vessels and serve as the boundary between blood and parenchymal cells, thus playing a crucial role in preserving organ function [14]. Therefore, ECs act as the first line of defense against invading pathogens. They possess pattern recognition receptors on their surface, as well as in immune cells, tissue macrophages, and monocytes. Activation of these receptors, often triggered by components of bacterial walls like LPS, initiates inflammatory and coagulation cascades. Endothelial activation is particularly beneficial in cases of local infection as it helps contain the pathogens and effectively controls the spread of infection [1,112,113].

The degree of endothelial permeability varies significantly among different organs and systems, determining their functional peculiarities. Specifically, the endothelial barrier is tighter in the central nervous system compared to the cardiac, skeletal muscle, or lung systems and even more so compared to the presence of discontinuous capillaries that allow blood cells to cross the endothelial barrier in the spleen or bone marrow [110,111]. Therefore, the specific qualitative and quantitative organ damage depends on the direct or indirect insult it is subjected to, the integrity of the directly or indirectly insulted endothelium, the presence and tightness of tight junctions, the number of receptors expressed in the involved organs, and consequently, the secondary expression and release of ADM.

Subsequently, widespread loss of endothelial barrier integrity that can occur during sepsis or cytokine storm during COVID-19 results in increased endothelial permeability syndrome (GIPS), with edema formation inducing pulmonary edema and systemic hypotension that, leading to organ hypoperfusion, causes organ failure [1,114].

The functions of the EC lining are further reinforced by the glycocalyx, which regulates blood clotting, white blood cells and platelet adhesion, transfer of shear stress to the endothelium [12], and defense against inflammation [14].

Changes in the lining of EC cells can be beneficial or harmful, depending on the severity of the disease, the duration since its onset, persistent bacterial or viral spread, and the host’s genetic background, which determines the extent of immune metabolic-inflammatory damage. Initially, a localized widening of blood vessels allows white blood cells to reach the site of the infection, while activation of blood clotting helps limit the spread of the infection. However, in later stages, these changes can lead to a septic phenotype, causing a reduction in overall blood vessel tone, increased fluid loss from blood vessels, impaired blood flow in smaller blood vessels, and abnormalities in blood clotting, which can cause disseminated lesions and intravascular coagulopathy [115,116].

Moreover, venous thromboembolism is strongly associated with vascular endothelial impairment and a heightened tendency for blood clot formation [117,118].

Briefly, a noxa (viral or bacterial infection) can cause damage to ECs, resulting in dysregulation of the ADM system, leading to dysfunction of the activities carried out physiologically by the system, as shown in Figure 2A. This leads to (Figure 2B) a marked reduction in the vasodilatory effect and increases in blood flow, a loss of the integrity of the ECs with reduced vascular permeability, inhibition of the RAAS system and the catecholamine (and insulin) production [1,69,70,71] with reduced vasoconstriction effect, a loss of immunoregulatory property with reduced transcription of proinflammatory genes and increased transcription of anti-inflammatory cytokines, as well as a loss of antimicrobial properties leading to loss of the immune barrier, and inhibition of bacterial and fungal growth [1,11,50,58]. These alterations cause arterial hypotension, increased inflammatory and infectious status, and vascular leaks. This vascular leakage leads to an increase in the inflammatory state and the activation of the coagulation cascade, which affects various organs. Specifically, at the level of the blood vessels, an increase in vasoconstriction, a proliferation of VSMCs, an increase in apoptosis, a reduction in angiogenesis and an increase in endothelin production are triggered. At the cardiac level, a reduction in coronary flow and ANP production, cardiac hypertrophy, and an increase in myocardial fibrosis are established [1,9,13]. At the pulmonary level, there is vasoconstriction of arterioles, which has a profibrotic effect [1,69,70,71]. An increase in aldosterone production is established at the level of the adrenal glands. At the level of the central nervous system, a reduction in vascular flow and an increase in salt and water intake requirements, as well as ACTH and arginine vasopressin (AVP) secretion, is established. At the kidney level, a reduction in natriuretic and diuretic effects and an increase in mesangial cell production is established [1,9,13]. Furthermore, a counterregulatory action is put in place against the RAAS system [5,6,73]. These effects result in reduced cardiac output, myocardial injury and ischemia, increased vascular wall resistance, subendocardial damage of the right ventricle, endothelitis, microvessel permeability, and thrombus formation [4,5,16]. On the other hand (Figure 2C), the hypoxia that occurs during a severe infection determines a teleological protective increase in expression of ADM both through Connexin C43 -which stimulates lymphangiogenesis (reducing edema and fibrosis and, therefore, myocardial remodeling), both through Akt-GSK caspase, which reduces apoptosis and increases cell viability of myocardiocytes [4,7,73,83,84,85,86,87,88,89,90,91,95,119].

This evidence underscores the crucial functions of ADM and the importance of information on the degree of endothelitis and dysregulation of the ADM system provided by the ADM and/or MR-proADM assay [4,120,121,122,123]. Indeed, the value of MR-proADM, which correlates directly with the extent of organ damage, indicates organ dysfunction, disease severity, diagnosis and prognosis in patients with severe viral and bacterial infections [4,120,121,122,123]. 

Furthermore, MR-proADM acts as a biomarker for sepsis, reflecting the degree of oxidative stress and indicating the severity of the condition in line with organ damage and prognosis [120,124]. From a clinical perspective, MR-pro-ADM can accurately detect patients with one or multiple organ failures, particularly showing a stronger correlation for certain variables such as acute kidney injury (AKI), GCS < 15, and the requirement for intensive care unit (ICU) transfer indicating a more severe prognosis [19,20,120,125,126,127,128,129].

The primary purpose of utilizing MR-proADM in clinical practice is to determine the specific characteristics of septic patients who are at higher susceptibility to life-threatening organ impairment and mortality. Thus, it helps identify patients who require prompt and intensive therapeutic intervention.

In a recent study, it was found that the designated threshold level of MR-proADM (which achieved the optimal balance between sensitivity and specificity) for detecting patients with localized infection was 1.44 ng/mL. For identifying patients with sepsis with a SOFA score greater than 2, as well as distinguishing those needing ICU transfer or unlikely to survive beyond 90 days, the threshold value proved to be 2 ng/mL [120].

Furthermore, MR-proADM showed predictive power for mortality in septic patients, independent of the extent of organ dysfunction [129].

## 6. Pathophysiology of Severe Viral and Bacterial Infections and the Immune Modulation in the Intravascular Compartment

As for the mechanism by which viral and bacterial pathogens cause damage to the intravascular compartment, much has yet to be defined. It appears that both of these groups of pathogens can adhere to the cell surface through the angiotensin-converting enzyme 2 receptor (ACE2r), which causes their entry into the cell [130].

It will be considered how viral and/or bacterial infection cause damage in the intravascular compartment (Figure 3) and direct injury to ECs of myocardiocytes and perimycytes (Figure 4). Specifically, in the case of SARS-CoV-2 infection (used as the most studied example of viral infection), the virus adheres to the cell surface through the angiotensin-converting enzyme 2 (ACE2)-receptor, which causes it to enter the cell [130].

Bacterial infections can also cause damage to the ACE2 system localized at the level of ECs, but further studies are needed to investigate the pathophysiological mechanisms (Figure 3). In experimental studies in mice, sepsis induces downregulation of ACE2 by inhibiting mitochondrial synthesis and function, resulting in myocardial injury with organ failure [131].

Inside the cell, Toll-like Receptors (TLR3, TLR4, TLR7 in viral infection, or TLR4 in bacterial infection) determine (a) the release of danger-associated molecular patterns, (b) the activation of the inflammatory response, and (c) the activation of innate pathways. All of this results in viral infection in the release of viral RiboNucleic Acid (RNA) into the cell up to the exocellular release of the RNA genome, which is copied and attached to the nucleocapsid protein [132]. Outside the cell, (1) B lymphocyte receptors, in viral infection, recognize the spike glycoprotein and nucleocapsid protein, while B cells produce spike glycoprotein-binding antibodies and neutralizing antibodies. Instead, in bacterial infection, they phagocytize the pathogen and form the phagolysosome by presenting with major histocompatibility complex class II (MCH II) on the cell surface digested antigens [133] (2) antigen-presenting cells (APCs): (a) activate Cytotoxic T Lymphocyte (CTL) which kill infected cell (b) present the antigens to the Helper T cells (Th cells), which become activated and produce cytokines (mainly Interferon (INF)-γ, Interleukin (IL)-2, TNF-α; (c) activate, recognize and kill the infected cells [133,134].

Then, in case of both viral and/or bacterial infection, monocytes and T cells release -until the exhaustion of lymphocytes and T-cell deficiency- tumor necrosis factor ligand superfamily member 14 (TNFSF14: Light), which determines an increase of autoantibodies anti-AT1 receptor and, probably, also antiACE2, increasing activity pro-inflammatory and fibrotic of RAAS, activation of intrinsic and extrinsic pathways of coagulation with platelet hyperactivation and hyperaggregability, and release of cytokines (Interleukin (IL)-1β, IL-18, platelet-derived extracellular vesicles (PEVs), inactivation of natural pathways of anticoagulation Tissue factor (TF) pathway inhibitor I (TFPI) and Thrombomodulin through activation of monocytes, neutrophils, activation of the Complement (C3–C5) pathway resulting in pathogen killing, containment of organ damage, hypercoagulability, platelet hyperaggregation, fibrosis and vasoconstriction with hypoxia [132,134]. Specifically, via activation of pathogen recognition receptor (PPR), monocytes activate the extrinsic coagulation pathway, TNFSF14 and NLR family pyrin domain containing 3 (NLRP3), stimulating the release of IL-1β and IL-18. Tissue factor pathway inhibitor I (TFPI) is the physiological inhibitor of TF-induced blood coagulation. Neutrophils, by activating NETosis, activate factor XII—which activates the contact-dependent coagulation pathway-, bind Von Willebrand factor—which recruits platelets (PLTs), resulting in increased PLTs and activation of EC (which goes to enhance activation of contact-dependent coagulation pathway) with release of fibrin that goes to trap the pathogen killing it and increased PEVs-, activates Histones -which by releasing H3 and H4 activate PLTs-, activates Neutrophil elastase (NE) and myeloperoxidase (MPO) which cleave and inactivate the natural anticoagulation pathway TFPI and thrombomodulin-, and expresses and binds TF which activates the extrinsic coagulation pathway [132,134].

Furthermore, the direct injury of myocardial ECs and perimyocytes induced by SARS-CoV-2 is also aligned with the damage induced by sepsis, which is explained in Figure 4.

## 7. Pathophysiology of Severe Viral and Bacterial Infections on Endothelial Cells: Endothelitis and ADM Expression

Severe viral or bacterial infections cause inflammation with increment of C-reactive protein (CRP) and neopterin and oxidative stress with increased production of reactive oxygen species (ROS) and reactive nitrogen species (RNS) and subsequent increment of NOx levels [22,23,45,133]. This results in direct damage to ECs (endothelitis with an increase of ADM expression), of the junctional proteins (Jps) between ECs and of the glycocalyx of ECs, resulting in impairment of the ACE2 and ADM systems, and of the Angiopoietin/Tie (Ang/Tie) axis [45,120,132,133]. 

ACE2 is the entry receptor for the noxa, and upregulation of ACE2 expression enhances cell infection [120,132]. The pathogen binding ACE2 determines the reduction of the level of ACE2 on the cell surface of monocytes, ECs, myocardiocytes, perimycites, pulmonary cells and VSMCs. This determines a shift in RAAS balance towards the proinflammatory Angiotensin II/AT 1 axis, which leads to inflammation, fibrosis, and progression of disease severity [120]. This determine an increase in RAAS activity and vascular leakage, an increase in inflammatory and coagulative triggering-cascade, and lymphangiogenesis resulting in vasoconstriction, increased permeability, edema, inflammation, oxidation, proliferation, fibrosis (Figure 4) [130,135,136,137].

Specifically, ACE2 transforms proinflammatory components of the RAAS axis (angiotensin I and II) into anti-inflammatory components such as Ang 1-9 and especially Ang 1-7 [135]. Ang 2 acts on angiotensin I (AT1 receptors) and activates the NAPDH-oxidase complex, producing superoxide and promoting cell-oxidative and proinflammatory responses, while Ang 1-7 acts on MAS and mAS-related receptors promoting cell antioxidative and anti-inflammatory response severity [130,135,136,137].

Autoantibodies against major components of RAAS (autoantibodies for AT1 receptor -AA-AT1-), which act as AT1 receptor agonists, enhance the proinflammatory RAAS activity in several tissues, and process severity [130].

Autoantibodies anti-ACE2 act by inhibiting ACE2 function, reducing the anti-inflammatory activity, shifting the balance towards the proinflammatory RAAS severity [130,138].

Increased cytokine levels (IL-6, TNFα, especially TNFSF14) induce the generation of anti-AT1 autoantibodies (AA-AT1), determining the severity of the infection severity [130,138].

The pathological conditions that occur during severe viral and bacterial infections, such as hypoxia, hypo or hypervolemia, hyperglycemia, tissue ischemia [5,27,43,137,139], bacterial endotoxin -LPS- [59], ROS, proinflammatory cytokines -TNF-α-and TGF-β [43,59], IL-1β, INF -γ-, substance P, endothelin-1, bradykinin, norepinephrine, corticosteroid and thyroid hormones, angiotensin II, shear stress and stretching [5,134,138,140,141,142,143,144,145,146,147,148], determine inflammation and oxidative stress responsible for damage to cells of the immune system, endothelitis, with damage of the endothelial wall -consisting of a single layer of ECs, sealed by intercellular junction and covered by the extracellular intraluminal structure glycocalyx-, resulting in biohumoral dysregulation, and increase of expression of ADM, as is showed in Figure 4 [149].

In addition, the stability of the endothelial barrier is determined by the integrity and functionality of ECs and intracellular junction molecule expression. The insult of the pathogen (viral or bacterial) or the oxidative or wall stress leads to the loss of barrier immunity by the Ecs, allowing damage and exposure of the endothelial glycocalyx, vascular leakage up to interstitial edema and shock [5,150]. Specifically, degradation of endothelial glycocalyx results in (a) loss of barrier immunity; (b) increased permeability to proteins -including albumin- and to free water; (c) exposure of adhesion molecules on the surface of ECs activating neutrophils contributing to microvascular thrombosis; (d) increased diameter of microscopic vessels with secondary hypoperfusion of the other capillary beds; (e) loss of shear stress monitoring and signaling from the glycocalyx structure to the ECs and VSMCs, contributing to the loss of vascular reactivity; (f) increase in the spread of ADM from intravascular to interstitial with an increase in vasodilatation, passage of albumin into the interstitial space and therefore increase in vascular leakage, systemic and pulmonary edema, hypotension, hypoperfusion and organ dysfunction, shock and death [5,150]. Furthermore, dysregulation of the specific endothelial receptors that regulate the contraction of Jps (the Ang/Tie axis) leads to inhibition of Jps, destabilization of cortical actin, increase in vascular adhesion and permeability, formation of inflammatory thrombi, activation of the coagulation causing microcirculatory dysfunction and organ failure (Figure 4) [151]. Specifically, the correct functioning signals of the tight junctions occur thanks to the formation of a heterodimer formed by the union of Angiopoietin with the specific Tie receptors (Ang 1 with Tie 1 and Ang2 with Tie2, respectively) [5,151]. As a result of this union, Tie 2 triggers cortical actin formation and upregulates pathways related to the anti-adhesion and anti-inflammatory properties of ECs, maintaining vascular stability. Ang/Tie2 activation induces KLF2 expression through the PI3K/Akt pathway and counteracts with the vascular endothelial growth factor (VEGF)-mediated vascular permeability, while elevation of intracellular NO by endothelial NOS (eNOS) expression is potent to inhibit exocytosis of Ang2 from Weibel–Palade bodies (WPB) found in endothelial cell cytoplasm [148,152,153].

In the Angiopoietin family, Ang1 acts as an agonist, while Ang2 is predominantly an antagonist. Elevated Ang2 levels correlate with the severity of ARDS and sepsis [154,155]. Also depicted in Figure 4 are the process steps in which the inhibitors act: angiotensin-converting enzyme inhibitor (ACEi) and angiotensin receptor blocker (ARB) on the RAAS) [135], adrecizumab on the ADM system [156], and Ang2-binding antibody (ABTAA) on the Ang/Tie axis [151].

## 8. Role of Adrenomedullin Expression in Severe Viral and Bacterial Pneumonia

The relatively recent use of biomarkers such as ADM in infections has allowed only a few studies in influenza-type viral lung infections and prevalent use in SARS-COV-2 infections.

The first studies of MR-proADM plasma levels in patients with Influenza A virus pneumonia showed that MR-proADM could predict disease severity, unfavorable outcome, risk of ICU admission, need for mechanical ventilation and mortality. Moreover, the prognostic superiority of MR-proADM levels over other markers such as PCT, CRP, and ferritin has been demonstrated, as well as the severity scoring systems SOFA and APACHE II. Then, MR-proADM stratifies high-risk patients adequately and is likely to benefit new treatments in patients with H1N1vIPN [157]. In particular, values of MR-proADM above 1.1 nmol/L at hospital admission identify patients with H1N1vIPN who developed severe respiratory failure with the need for mechanical ventilation in a high percentage of cases. The use of MR-values above 1.2 nmol/L detects poor-prognosis ICU patients with H1N1vIPN, indicating 100% mortality, 100% sensitivity, and a negative predictive value [157,158,159].

COVID-19 provides an example of a systemic disease that compromises widespread endothelial damage with multiple organ dysfunction syndrome in severe cases and resulting increase in ADM expression [14,160].

MR-proADM, therefore, represents a marker of endothelitis predicting COVID-19 severity [55,161]. In COVID-19 patients, MR-proADM stratifies patients with a worse prognosis at risk of major organ damage, including ARDS, with a greater need for oxygen therapy, at risk of ICU transfer, in which monoclonal antibody (Adrecizumab) therapy may be efficacy [14,156,162].

In this regard, MR-proADM levels were an independent predictor of 90-day mortality, showing a cutoff of 0.80 nmol/L, a sensitivity of 96.9% and a specificity of 58.4%, with a negative predictive value of 99.5% [8,55]. In addition, MR-proADM values ≥ 2 nmol/L identify patients with high mortality risk related to a multiorgan dysfunction syndrome, while ≥ 3.04 nmol/L identify patients who will develop ARDS. Such patients should also be considered for an intensive therapeutic approach, including monoclonal antibodies [14].

Bacterial coinfections and ischemic-thromboembolic complications worsen the outcome and prognosis of COVID-19 patients [163].

MR-proADM demonstrated a strong capability to detect bacterial pneumonia, particularly in patients with more severe and complex clinical conditions. This finding was further supported by a significant and positive correlation between MR-proADM and severity index scores, such as pneumonia severity index (PSI) score values. This correlation solely represents the severity of pneumonia [164]. The same conclusions were confirmed by a study of patients with community-acquired pneumonia (CAP) in the emergency department (ED), in which the highest MR-proADM values were found in patients with more severe pneumonia (multilobar rather than unilateral), positive blood culture, or who passed away (2.1 vs. 1.0 nmol/L). The MR-proADM value of 1.8 nmol/L showed a sensitivity of 80% and specificity of 72%, a positive likelihood ratio of 2.9 and a negative likelihood ratio of 0.28 in prognosticating one-year mortality. The researchers conclude that the MR-proADM level on admission predicts the severity and outcome of the CAP patient and can support the role of PSI in therapeutic or hospitalization decision-making, showing itself to be a useful tool for risk stratification of the CAP patient [165]. Therefore, MR-proADM successfully stratifies patients with both viral and bacterial severe lung infections.

## 9. Role of Adrenomedullin Expression in Infective Acute Cardiac Injury

Heart involvement (inflammatory, ischemic, thromboembolic, and arrhythmic) during viral infections is much more represented than the known prevalence. Indeed, not all patients with influenza or SARS-CoV-2 infection, bacterial pneumonia, or sepsis perform electrocardiograms, echocardiograms, or level III imaging methods. Specifically, in critically ill patients with viral infections, particularly with SARS-CoV-2, acute cardiac injury is found in up to 15–50% of cases, represented by myocardial injury, endothelitis, heart failure, Takotsubo cardiomyopathy, acute coronary syndromes, pulmonary thromboembolism, and arrhythmias [163,166,167,168,169,170,171].

The high rate of acute cardiac injury caused by SARS-CoV-2 is similar to that observed in other viral infections, such as influenza, where myocardial damage was found in 0–53% of cases without any symptoms of cardiac involvement, and about 50% of patients had an abnormal electrocardiogram. This damage was also detected during postmortem examinations through the finding of myocarditis, pericarditis, or acute coronary syndrome [163,166,167,168,169,170,171,172,173,174]. 

In addition, a very interesting retrospective study showed an increase in myocardial ^18^F-FDG uptake at PET/CT in patients who underwent this method of imaging for noninfectious indications other than myocardial inflammation after receiving the second dose of anti-SARS-CoV-2 vaccine mRNA within 180 days, compared with unvaccinated patients. Unfortunately, a correlation between the increase in myocardial ^18^F-FDG uptake and the presence of symptoms, clinical or instrumental signs, or biohumoral examinations that are suggestive of ongoing SARS-CoV-2 myocarditis or other infections is unknown [175]. 

Myocardial injury is defined as an increase in myocardial enzyme levels (Troponin) with at least one value above the 99th percentile upper reference limit in the absence of myocardial ischemia and can be caused by several mechanisms [115]. Myocardial injury occurs due to indirect or direct myocardial damage [169]. Indirect myocardial injury evidenced by increased Troponin is present up to 36% in the early course of SARS-CoV-2 infection and is associated with an increased risk of requiring mechanical ventilation, fatal ventricular arrhythmias, and a 59.6% risk mortality [176,177,178,179,180,181]. A direct myocardial injury affects high sensitivity (hs) Troponin I in case of acute coronary syndrome. It could affect the expression of ADM, which is expressed by cardiomyocytes, pericytes, cardio-fibroblasts, ECs, epicardial adipose cells, VSMCs, and migratory angiogenic cells [170,171].

Myocardial damage might occur during SARS-CoV-2 infection as a consequence of myocardial, pulmonary, and endothelial damage. It is due to hypoxia inducing a reduced oxygen supply to the heart, causing a modest or massive elevation in troponin concentration, which is not necessarily related to the deterioration of systolic left ventricular function in left ventricular overload secondary to parenchymal or vascular disease of the lung resulting in subendocardial damage of the right ventricular myocardium in 19% of cases and by cytokine-induced injury [177,178,182,183,184,185,186]. Myocardial injury can be caused through one of three mechanisms: myocyte stretching, myocardial damage, or oxidative stress [187]. The MR-proADM biomarker indicates cardiovascular stress [187,188].

A recent study demonstrated how MR-proAM complements troponin, a canonical biomarker of myocardial damage, improving its prognosis accuracy and risk stratification in a cohort of COVID-19 patients with myocardial injury [187,189]. The elevation of hs Troponin I and MR-proADM allows the identification of patients with myocardial injury at higher mortality risk [190]. An MR-proADM value of ≥1.19 nmol/L identifies patients with myocardial injury, and an MR-proADM value of ≥4.01 nmol/L identifies patients with myocardial injury at high risk of death [187]. In patients with myocardial injury, the mortality rate was significantly higher in those with elevated values of both hs Troponin I and MR-proADM ≥ 1.19 nmol/L, reaching 53.2%, compared to a mortality rate of 14.8% in patients with elevated values of hs Troponin I only. Moreover, the presence of elevated levels of both biomarkers helped identify individuals with myocardial injury who were at a greater risk of mortality. When both biomarkers were negative, the mortality rate was only 4%, but when both were positive, the mortality rate increased to 53.2%. Therefore, measuring MR-proADM levels can help stratify patients with myocardial injury and identify those who may benefit from adrecizumab therapy [187]. Indeed, viral infections have the ability to induce endothelial dysfunction, which can ultimately lead to apoptosis and trigger coronary vasoconstriction. Additionally, they can cause a procoagulant state, leading to plaque activation and hemodynamic instability [159,191].

In the context of bacterial infections, infective endocarditis (IE) should be considered separately. A recent, very interesting retrospective study showed that in patients with IE, an MR-proADM value > 1.05 nmol/L is an independent predictor of in-hospital mortality with a sensitivity of about 80%. Higher MR-proADM values were found in left heart IEs than in right heart IEs and in IEs sustained by *Enterococci* and *Staphylococci* [192].

However, sepsis-related cardiomyopathy is common in up to 38–64% of severe bacterial infections [193,194]. The definition of sepsis-induced myocardial dysfunction indicates left ventricular (LV) systolic dysfunction. In contrast, both ventricles can be impaired and when right ventricular (RV) systolic dysfunction is present, the prognosis of septic patients is poor [193,194,195,196]. In septic patients, two types of LV dysfunction were represented: typical septic cardiomyopathy (SC) and sepsis-related Takotsubo cardiomyopathy (ST). In ST patients, LV apical and circumferential mid-ventricular hypokinesia and basal hypercontractility usually occur [196]. In a recent study involving septic patients, acute heart failure (AHF) needing the use of catecholamines was present in 33.5% of the cases [120]. In septic patients, the MR-proADM values at admission ≥ 2.28 ng/mL correlated with AHF requiring catecholamine administration correlated significantly with death at 30 days and multi-organ damage [19,120].

## 10. Role of Adrenomedullin in Organ Damage and Patients’ Prognosis

Since 2016, MR-proADM has been shown as a biomarker of organ damage in septic patients. Indeed, MR-proADM expresses the severity of the failure of the involved organ and/or the multiplicity of compromised organs, regardless of the infectious etiology [197].

MR-proADM indicates the severity of the patient, directing therapy and the need for hospitalization (in the case of a patient in the ED) or the most appropriate hospitalization setting (medical or intensive care unit) [198].

MR-proADM serves as a reliable marker for organ dysfunction in both individuals with localized infection and those suffering from sepsis and/or septic shock, using different cut-off values. Research revealed that AKI, which is strongly linked to elevated MR-proADM levels, is the most significant type of organ damage observed in infected and septic patients. Furthermore, septic patients also show obvious brain damage concomitant with AKI. Among septic patients, an MR-proADM cut-off of ≥2 or ≥3 ng/mL is strongly associated with several clinical parameters. These include AKI, anemia, AHF, a Glasgow Coma Scale (GCS) score below 15, the need for ICU transfer, the need for catecholamine administration, a SOFA score of ≥2, and multiple organ damage and shock. However, no significant correlation was observed with acute liver failure or a q-SOFA score of ≥2 [120].

MR-proADM has also been shown to be a valid marker of organ damage in stroke-associated pneumonia (SAP) cases, playing a crucial role in predicting the risk of pneumonia in stroke patients with a score NIH Stroke/Scale equal to 10 or higher due to its early elevation (24 h after stroke onset) [199,200]. Furthermore, in the presence of chronic obstructive pulmonary disease, pulmonary embolism or acute heart failure, MR-proADM dosing increases prognostic accuracy [200,201].

Noteworthy is the occurrence of post-surgical complications of both infectious and hemodynamic nature, where MR-proADM demonstrates its usefulness [25,202,203]. 

A preoperative MR-proADM level ≥ 0.70 nmol/L has been shown to be an independent risk factor for postoperative organ support with a negative predictive value of 91% [202]. Therefore, MR-proADM may be a valid biomarker for perioperative risk assessment [202].

A study from 2018 assessed the usefulness of preoperative MR-proADM and highly sensitive Troponin T in identifying patients with a heightened risk of experiencing cardiac events and mortality following significant noncardiac surgery. Compared to highly sensitive troponin T, MR-proADM proved to be a more reliable predictor of cardiovascular complications during the perioperative period [203].

A recent study investigated the roles of MR-proADM in AKI patients after cardiac surgery. The results showed that in septic patients, MR-proADM levels were measured at 2.3 nmol/L (0.7–7.8 nmol/L), with the highest levels observed in septic shock patients (5.6 nmol/L (3.2–18 nmol/L) and MR-proADM showed a better diagnostic profile compared to PCT and CRP in identifying these patients. Moreover, MR-proADM values > 5.1 nmol/L were associated with a significantly faster progression to renal replacement therapy (RRT). Therefore, MR-proADM provides early diagnosis of AKI in septic patients after cardiac surgery, offering prognostic information for the need for RRT [202]. 

MR-proADM would not show itself as an indicator of anaerobic metabolism if it were to weakly correlate with lactates [204]. Zelniker et al. studied the relationship between MR-proADM, lactate, and the risk of death in patients who experienced out-of-hospital cardiac arrest. The main outcome of interest was the occurrence of death from any cause. Patients were divided into four groups based on MR-proADM and lactate levels. Median MR-proADM and lactate levels at 24 h after the arrest were 1.4 nmol/L and 1.8 mmol/L, respectively. There was a weak correlation between MR-proADM and lactate levels. Patients with high MR-proADM levels had a significantly higher rate of death at 28 days than those with low MR-proADM levels. This relationship remained significant even after accounting for other biomarkers, such as NT-proBNP and troponin-T levels. Furthermore, patients with high levels of both MR-proADM and lactate had a particularly high risk of death. In conclusion, higher MR-proADM concentrations are associated with an increased risk of death in patients with out-of-hospital cardiac arrest. Additionally, the combination of high MR-proADM and lactate levels identifies patients who are at an even higher risk of death [204].

A progressive reduction of MR-proADM from day 2 to day 5 of hospitalization in a critical septic patient, indicating response to therapy, is a favorable prognostic factor [197,205]. MR-proADM has also proven useful as an indicator of hospital readmission and mortality. Measurement of MR-proADM in patients with multimorbidity may be useful in predicting the risk of readmission and/or death within 90 days of admission to the ED. MR-proADM > 0.75 pmol/L was significantly associated with a higher risk of readmission and/or death. Moreover, adding MR-proADM to age, sex, and multimorbidity improved the predictive value [206].

The early detection and appropriate treatment of septic patients are crucial in reducing the risk of mortality. Current risk stratification for patients with infection is typically determined using the SOFA score or NEWS.

A study involving patients with severe sepsis or septic shock admitted to an internal medicine ward, aimed to investigate the relationship between biomarkers and mortality at 90 days, showed that MR-proADM exhibited the highest predictive ability for sepsis, according to the Sepsis-3 criteria, and the best biomarker to independently predict 90-day mortality. The cut-off-point of MR-proADM of 1.8 nmol/L demonstrated the greatest discriminative capacity to predict mortality. Remarkably, all patients with MR-proADM concentrations ≤ 0.60 nmol/L survived up to 90 days. Additionally, in patients with SOFA ≤ 6, the addition of MR-proADM to SOFA score increased the ability of SOFA to identify non-survivors. In conclusion, MR-proADM resulted in a good biomarker in the early identification of high-risk septic patients and could enhance the predictive ability of the SOFA scale, especially when scores are low [207].

However, current biomarkers and low SOFA scores may not accurately identify patients who may develop severe organ dysfunction or have a higher risk of mortality. A recent study aimed to evaluate the predictive value of biomarkers such as MR-proADM, PCT, CRP and lactate for 28-day mortality in septic patients with a SOFA score of less than 6. Lactate showed the highest predictive ability for all-cause 28-day mortality. In patients with community-acquired infection, a cut-off point of 2.1 nmol/L for MR-proADM in this subgroup of patients accurately identified survivors from non-survivors at 28 days with 100% sensitivity. Therefore, MR-proADM may help identify patients at risk of 28-day mortality in patients with community-acquired sepsis and an initial SOFA score of less than 6 [208].

Andaluz-Ojeda et al. investigated the ability of MR-proADM to predict the likelihood of death in sepsis patients with different degrees of organ failure, compared to PCT, CRP, and lactate, within the first 12 h of admission. In the multivariate analysis, only MR-proADM and lactate were associated with mortality. MR-proADM was the only biomarker that predicted mortality in all severity groups, and all patients with MR-proADM concentrations ≤ 0.88 nmol/L survived up to 28 days. Moreover, in patients with SOFA ≤ 6, adding MR-proADM to the SOFA score increased the ability to identify non-survivors. These findings suggested that the accuracy of MR-proADM in predicting mortality is not only influenced by the degree of organ failure, making it a promising tool for early identification of sepsis patients with moderate disease severity who are at risk of dying [129].

In a recent study, MR-proADM was analyzed to determine whether it could improve the predictive value of the SOFA score for 30-day mortality in acutely infected patients attending the ED. SOFA score, MR-proADM and traditional markers of inflammation were measured at the time of admission.

It was found that non-survivors had higher MR-proADM levels compared to survivors (4.5 ± 3.5 nmol/L vs. 1.7 ± 1.8 nmol/L), with an adjusted odds ratio of 26.6 for every 1 nmol/L increase in admission MR-proADM levels. An admission MR-proADM threshold of 1.75 nmol/L was identified as the most accurate predictor for 30-day mortality, with a sensitivity of 81%, specificity of 75%, and negative predictive value of 98%. These results suggested that MR-proADM improved mortality risk stratification in infected patients who attended the ED, in addition to the use of SOFA score alone, indicating that MR-proADM may improve initial treatment decisions [209].

A prospective observational study of patients undergoing sepsis code activation was conducted to identify appropriate combinations of biomarkers (MR-proADM, PCT, CRP, and lactate) or clinical scores (SOFA and APACHE II) to address this clinical need. Both PCT and MR-proADM showed moderate to high performance in distinguishing between infected and uninfected patients following sepsis code activation, although the optimal threshold of PCT varied among departments. Similarly, MR-proADM and SOFA were effective in predicting 28- and 90-day mortality among the infected patient population, as well as among those with community-acquired infections or those experiencing medical emergencies or prior surgical procedures. Importantly, MR-proADM was also strongly associated with the need for ICU admission after ED presentation or during ward treatment. The study suggested that individual use of PCT and MR-proADM could help accurately identify patients with infections and assess the severity of the host response, respectively. Additionally, MR-proADM could accurately identify patients who need ICU admission, regardless of the specific clinical setting [25].

Currently, there are few validated combinations of biomarkers or clinical scores that can effectively differentiate between cases of infection and noninfectious conditions following activation of a sepsis code in the hospital setting. In addition, these combinations must accurately assess the severity of the host response. The effectiveness of using four biomarkers (CRP, lactate, MR-proADM, and PCT) in identifying the various types of organ failure outlined in SOFA was evaluated in a cohort of patients presenting with infection, sepsis, or septic shock. On multivariate analysis, MR-proADM emerged as a distinct indicator for five distinct organ failures (respiratory, coagulative, cardiovascular, neurological, and renal). In contrast, lactate was predictive of three failures (coagulation, cardiovascular and neurological), while PCT indicated two failures (cardiovascular and renal). CRP showed no predictive ability for individual components of SOFA. None of the biomarkers examined were able to identify liver damage. Among patients with infection, MR-proADM stood out as the biomarker identifying the most components of the SOFA score, with the exception of liver failure [19]. Anticipating the onset or exacerbation of organ dysfunction in infected patients, identifying the most fragile patients, and administering timely antibiotics and supportive therapy would be extremely valuable. MR-proADM successfully predicted the incidence and deterioration of organ failure among critically ill patients, regardless of infection. Including diagnostic biomarkers of infection such as PCT, the progression of sepsis in infected patients could be predicted [210].

The use of biomarkers allows for an increase in the diagnostic and prognostic performance of clinical scores. Indeed, the use of NEWS and MR-proADM together predicted an increase in severity more accurately than the use of NEWS alone [201].

MR-proADM has been shown to be a potentially useful biomarker for predicting worsening in patients admitted to hospital with mild to moderately severe acute illness, particularly those with a NEWS score between 2 and 5, suggesting exploring the practicality and utility of developing a decision aid based on admission NEWS, MR-proADM level and other clinical data and biomarkers to further improve prognostic accuracy [201].

Combining NEWS with three biomarkers (such as white blood cell count, PCT and MR-proADM) or adding MR-proADM alone significantly improves the prediction of ICU admission or all-cause mortality at 30 days (with a multivariate-adjusted odds ratio of 1.26), as was shown in an observational study conducted in three tertiary care centers in France, Switzerland and the United States [211].

Finally, an association of MR-proADM with (abdominal) obesity, selected adipokines, and biomarkers of subclinical inflammation was found. Specifically, MR-proADM was associated with all-cause and cardiovascular mortality even after adjusting for traditional cardiovascular risk factors, including BMI or waist circumference. MR-proADM was also associated with four out of seven examined adipokines (leptin, retinol-binding protein-4, chemerin, and adiponectin) and with five out of eleven examined biomarkers of subclinical inflammation (high-sensitivity CRP, interleukin-6, myeloperoxidase, interleukin-22, and interleukin-1 receptor antagonist) after multivariable adjustment and correction for multiple testing. However, the association of MR-proADM with mortality was independent of these parameters and suggested that future studies should investigate the role of IL-6 and further characteristics of subclinical inflammation in the association between MR-proADM and all-cause mortality [212]. This evidence correlates with previous studies in mice in which obesity was shown to be correlated with the presence in adipose tissue of macrophages that induce ADM and CGRP mRNA, suggesting an increased role of inflammation in human obesity due precisely to increased expression of these peptides [57].

The presence and degree of endothelitis determine the extent of organ failure and prognosis [4,120,121,122,123,197]. Noteworthy are the microvascular changes that play a role in the onset of organ damage in critically ill patients. The study of how MR-proADM can predict organ damage and its progression is still in its early stages. A recent preliminary observational study conducted on a small group of patients diagnosed with infection, sepsis or septic shock explored the relationship between MR-proADM and microvascular flow index (MFI). The study examined the association between MR-proADM clearance and microcirculation variables and the correlation between MR-proADM and SOFA score. MR-proADM showed no significant correlation with MFI at the time of ICU admission. However, a 20% or more clearance in the first 24 h was associated with better MFI. The researchers concluded that although there was no direct correlation between MR-proADM and MFI at the time of admission, there was a strong correlation between MR-proADM clearance, MFI and other microvascular factors [205].

Further studies would be needed to investigate fascinating scientific horizons that are not yet fully known.

## 11. Role of New Experimental Endothelial Barrier Stabilization Drugs Affecting ADM Expression

From the evidence reported so far about ADM expression following potentially fatal endothelial barrier damage, we infer the importance of preserving the integrity of the endothelial barrier and the need to develop drugs that can stabilize it despite damage established by the pathogenic noxa. A recently emerged drug with this goal is a monoclonal antibody (Adrecizumab) that has been used advantageously in small groups of critically ill patients with SARS-CoV-2 infection [213], sepsis [214], acute heart failure [215], or inflammatory bowel disease [139,213,214,215]. 

Adrecizumab, a 160 kDa humanized monoclonal antibody, acts by binding to ADM without blocking it and, thus, without neutralizing its effects, could improve vascular integrity, reduce tissue congestion and consequently improve clinical outcomes [213,216]. It is unable to diffuse freely from the bloodstream to surrounding tissues. When administered, it causes a dose-dependent increase in plasma ADM bound to the antibody. By binding to the N-terminal side of ADM, the monoclonal antibody transfers ADM from the interstitial space to vascular cells, thereby improving vascular integrity and preventing leakage. As a result, this mechanism effectively traps ADM in the bloodstream [213,214]. 

It also protects against proteolytic degradation of the N-terminal side, thereby increasing the half-life of ADM. Maintained in circulation, ADM protects the function of ECs through binding the ADM receptor formed by the CRLR/RAMP2/3 complex and subsequent stabilization of adhesion junctions and cytoskeleton, while vasodilator properties on VSMCs via the cAMP/PKA pathway in the interstitium may be reduced [217]. Furthermore, in preclinical studies in animals treated with adrecizumab, levels of angiopoietin 1 increased [156]. This peptide is an agonist of EC tight junction integrity and function.

In an enlightened early study of eight critically ill patients with COVID-19 and life-threatening ARDS, adrecizumab administration was followed by a favorable outcome [162,218].

In phase 2 AdrenOSS-2 trial, in the 300 septic shock patients with elevated ADM (>70 pg/mL) and less than 12 h after starting treatment with vasopressors for septic shock who received a single infusion of adrecizumab (2 or 4 mg/kg body weight) or placebo, the latter was safe and well tolerated and resulted in improved organ function and reduced mortality at day 28 from 28% to 24% [219,220]. 

More information is expected from a new study in which septic patients are separated from patients with septic shock (a phase 2b/3 study, ENCOURAGE) [214]. 

Further information is needed to determine the ideal time for administration of adrecizumab. During sepsis, the ideal time for administration of this monoclonal antibody should be before the establishment of endothelial barrier damage, and thus the passage of ADM into the interstitium could be indicated at the biohumoral level by the significant increase in PCT [1,217,221]. Indeed, as previously reported, ADM and PCT belong to the same family as CGRP, which also plays a role in inflammation [45,46,47].

A new alternative therapeutic strategy to adrecizumab for individualized treatment to protect the integrity of the vascular barrier during sepsis is to inhibit elevated levels of PCT [222,223]. Two therapeutic approaches could achieve this therapeutic effect. One is through intravenous sitagliptin, which inhibits dipeptidyl peptidase 4 (DPP4) by blocking PCT activation and establishing vascular barrier damage. The other is through the administration of olcegepant, which, by blocking CRLR/RAMP1, preserves the integrity of the endothelial barrier in murine sepsis.

In detail, in the plasma of patients with sepsis, PCT is present in two different forms: one is a peptide composed of 116 amino acids, and the other is a shortened variant of 114 amino acids [224,225]. Notably, enzymatic cleavage (from 116 to 114 amino acids) is facilitated by DPP4 through the removal of the two amino acids, alanine and proline, from the N-terminus [151]. This truncated form binds to the CRLR/RAMP1 complex on endothelial cells, causing phosphorylation of VE-cadherin and disruption of its assembly, leading to vascular leakage [151,223].

Furthermore, the actions of PCT affect the CGRP receptor, which consists of a heterodimer composed of CRLR and RAMP1 [223,226,227]. PCT has been found to activate the Src pathway, leading to the phosphorylation of VE-cadherin at tyrosine 685 and the separation of the adaptor protein p120 from VE-cadherin in ECs [151].

Recent data have shown that elevated PCT levels after major surgery are also indicative of organ dysfunction, signs of postoperative capillary leakage and increased fluid and vasopressor requirements, and suggest impaired microvascular integrity [223,228,229,230]. PCT concentrations of 10 ng/mL, commonly found in individuals with sepsis, trigger the development of intense pulmonary edema in healthy wild-type mice [225].

In mice with sepsis, administration of an antibody against the N-terminal side of PCT resulted in reduced lung inflammation and increased survival [222]. Thus, inhibiting procalcitonin’s actions by blocking DPP4 with sitagliptin or blocking CRLR/RAMP1 with olcegepant in murine polymicrobial sepsis specifically protects the integrity of the endothelial barrier.

Regarding the ideal timing of sitagliptin administration, in preclinical animal (mouse) studies, ideal intravenous administration of sitagliptin 6 h after the onset of polymicrobial sepsis disease promises outstanding results [217,221]. Currently, glyptins are only approved for oral administration, but because of the gastroparesis and alterations in oral bioavailability that occur in septic patients, studies have been conducted in healthy volunteers on intravenous administration of sitagliptin, which was well tolerated and showed protective effects on the vessels, heart, and kidneys [6,9,12,215,231,232,233,234,235,236].

Furthermore, some preclinical studies and still few clinical trials are set on the interference of angiopoietin-Tie2 signaling axis, ADM or vascular endothelial (VE-) cadherin precisely to limit endothelial hyperpermeability and its secondary complications [1,151]. Monoclonal antibody therapy ABTAA clusters the Tie2 receptor, bringing the axis from Tie2 antagonist to Tie2 agonist with benefits on endothelial barrier and endothelial glycocalyx integrity and survival [237]. Instead, administration of ACEi, ARB, or sacubitril/valsartan may play a role in counteracting the increase in RAAS activity brought about by ACE2, but the hypotensive effect and oral formulation do not allow timely administration of these drug classes survival [10,133]. The experimental administration of Angiotensin 1-7 in an experimental shock model found beneficial effects [136,137,238].

Even considering advanced futuristic treatments, it’s remarkable to note that vaccination remains the most effective strategy for preventing the development of severe viral illnesses in both adults and children. Efficacy rates are achieved of 50–60% with vaccines against Influenza and 94–95% with mRNA vaccines against SARS-CoV-2, respectively [239,240,241]. Vaccination is particularly beneficial for high-risk individuals such as those over 65 years of age, young children, those with existing health conditions, and those with compromised immune systems. Furthermore, vaccination could potentially prevent cardiovascular damage, reduce mortality rates, and prevent bacterial coinfections in accordance with antimicrobial stewardship directives [159,191,240,242].

## 12. Concluding Remarks and Future Perspectives

Regarding the evidence reported so far, we can conclude by asserting that ADM reflects the level of oxidative stress at the endothelial barrier and thus the degree of endothelitis, the prediction of organ damage, diagnosis and prognosis of severe viral and bacterial infections, and that from the clinical point of view, the integration of clinical signs, clinical scores and value of ADM identifies the phenotype of patients with severe viral and bacterial infections with worse prognosis. The use of such a biomarker thus allows stratification of patients requiring early and intensive management.

We propose the inclusion of MR-proADM as part of the panel of biomarkers needed for the diagnosis and management of critically ill patients. We also hope that experimental studies on endothelial permeability during systemic inflammation can be implemented, which is an essential, fascinating and not fully studied field.

## Figures and Tables

**Figure 1 cells-13-00892-f001:**
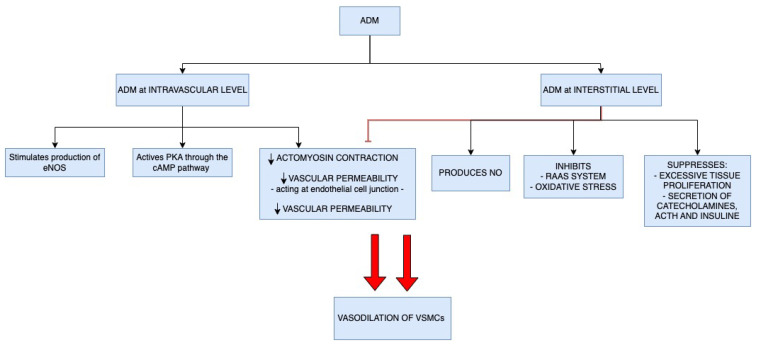
Vasodilator effect of adrenomedullin at the intravascular and interstitial levels. At the intravascular level, ADM induces vasodilation of VSMCs by stimulating the production of endothelial nitric oxide synthase (eNOS) and acting directly on these cells by activating protein kinase A (PKA). It also maintains vascular integrity by acting on endothelial cell junctions, reducing actomyosin contraction and vascular permeability. The latter function is inhibited by ADM at the interstitial level. In addition, at the interstitial level, ADM causes vasodilation through nitric oxide production, inhibition of the RAAS system and oxidative stress, and suppression of excessive tissue proliferation and secretion of catecholamines, ACTH, and insulin. Abbreviations: ACTH—Adrenocorticotropic hormone; ADM—adrenomedullin; cAMP—cyclic adenosine monophosphate; eNOS—nitric oxide synthase; NO—Nitric oxide; PKA—protein kinase A; RAAS—renin-angiotensin-aldosterone system; VSMCs—Vascular Smooth Muscle Cells.

**Figure 2 cells-13-00892-f002:**
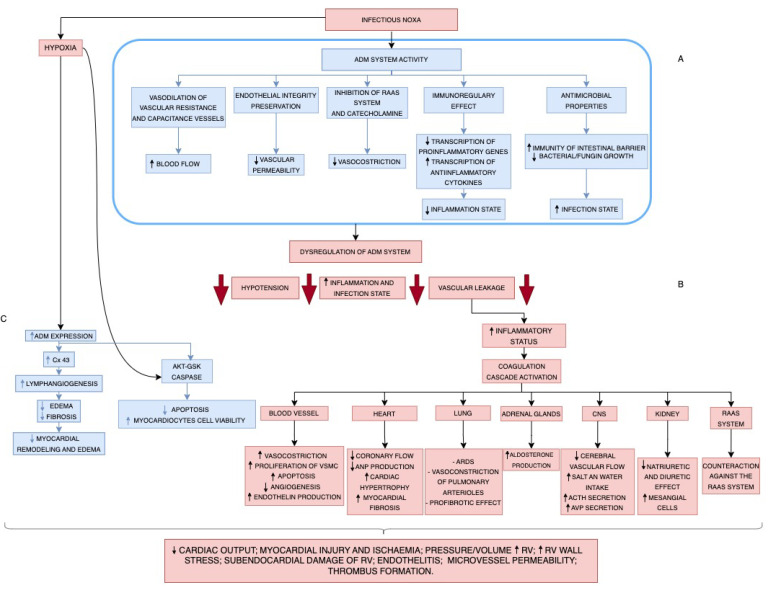
Pathophysiology of adrenomedullin expression: endothelitis. A noxa (viral or bacterial infection) can cause damage to ECs resulting in a dysregulation of the ADM system. This dysregulation leads to dysfunction of the activities guaranteed physiologically by the system, as shown in (**A**), light blue colored. This results in ((**B**) pink colored) hypotension, increased inflammation and infectious state, and vascular leakage leading to activation of the coagulation cascade and consequently to multiorgan failure. On the other hand ((**C**) light blue colored), the hypoxia that sets in with infection also results in increased ADM expression that teleologically stimulates lymphangiogenesis (with reduction of edema and fibrosis) and reduces cardiac remodeling, apoptosis and myocardiocyte viability. Abbreviations: ACTH—Adrenocorticotropic hormone; ADM—adrenomedullin; AVP—Arginine vasopressin; Cx43—Connexin 43; CNS—Central Nervous System; Endothelial cells (ECs); RAAS—renin-angiotensin-aldosterone system; RV—right ventricular; VSMCs—Vascular Smooth Muscle Cells.

**Figure 3 cells-13-00892-f003:**
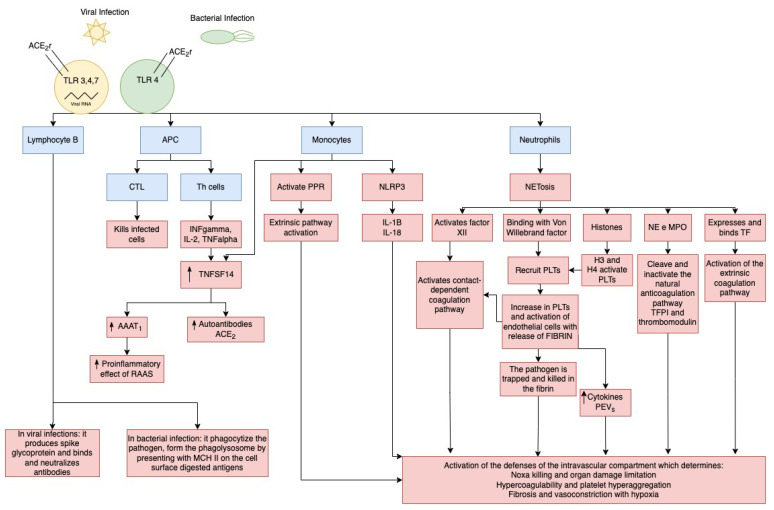
Pathophysiology of severe viral and bacterial infections and the immune modulation in the intravascular compartment. A viral or bacterial infection causes damage in the intravascular compartment by adhering to the cell surface through the ACE2 receptor, which causes it to enter the cell. Inside the cell, Toll-like Receptors and outside the cell B lymphocytes, antigen-presenting cells (APCs), monocytes, and neutrophils activate the immune response and the coagulation cascade. Abbreviations: AA-AT1—anti AT1 autoantibodies; ACE2—angiotensin-converting enzyme 2; ACE2r—angiotensin-converting enzyme 2 receptor; APCs—antigen-presenting cells; CTL—Cytotoxic T lymphocytes; H3—Histones 3; H4—Histone 4; IL-1B—Interleukin-1β; IL-2—Interleukin-2; IL-18—Interleukin-18; MHCII—major histocompatibility complex class II; MPO—Myeloperoxidase; NE—Neutrophil elastase; PEVs—platelet-derived extracellular vesicles; PLTs—Platelets; PPR—pathogen recognition receptor; RAAS—renin-angiotensin-aldosterone system; RNA—RiboNucleic Acid; TF—Tissue factor; TFPI—Tissue factor pathway inhibitor I; Th—Helper T cells; TLR 3,4,7—Toll-like Receptor 3,4,7; TLR 4—Toll-like Receptor 4; TNFα—Tumor necrosis factor α; TNFSF14 Light—tumor necrosis factor ligand superfamily member 14.

**Figure 4 cells-13-00892-f004:**
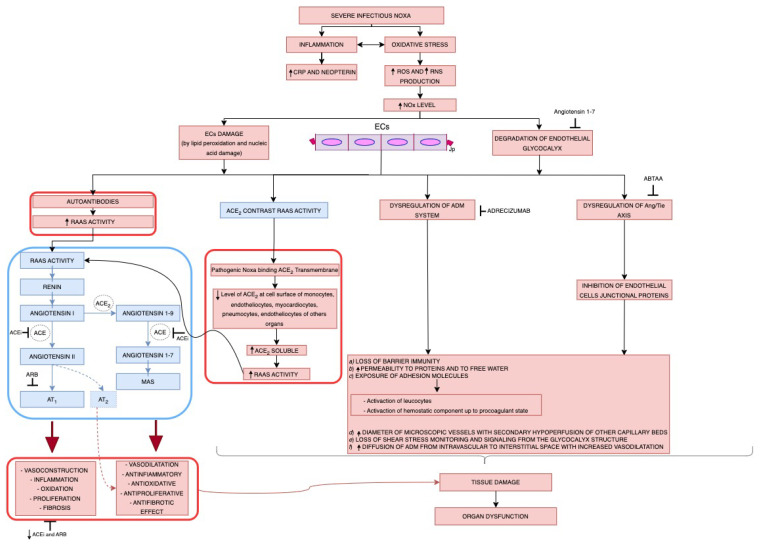
Pathophysiology of severe viral or bacterial infections on endothelial cells: endothelitis and ADM expression. Severe viral or bacterial infections cause inflammation and oxidative stress, leading to damage of Endothelial cells (ECs), degradation of glycocalyx with creation of autoantibodies resulting in increased RAAS activity, disruption of the ADM system and Angiopoietin-Tie axis with dysregulation of junctional proteins (Jps), activation of coagulation until organ failure sets in. Also depicted in the figure are the process steps in which the inhibitors act: ACE inhibitors and ARB on the RAAS, adrecizumab on the ADM system, and ABTAA on the Ang/Tie axis. Abbreviations: AA-AT1—anti AT1 autoantibodies; ABTAA—Ang2-binding antibody; ACE—angiotensin-converting enzyme; ACEi—angiotensin-converting enzyme inhibitor; ACE2—angiotensin-converting enzyme 2; ADM—adrenomedullin; Ang-Tie—Angiopoietin-Tie; Ang 1-7—Angiotensin-(1-7); ARB—angiotensin receptor blocker; ECs—Endothelial cells; CRP—C-Reactive Protein; NOx—Nitric oxide; RAAS—renin-angiotensin-aldosterone system; RNS—reactive nitrogen species; ROS—reactive oxygen species.

## Data Availability

Not applicable.

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
