# Peer review of "A Focus on the Pathophysiology of Adrenomedullin Expression: Endothelitis and Organ Damage in Severe Viral and Bacterial Infections"

_cells, 2024, doi:10.3390/cells13110892_

Round 1

Reviewer 1 Report

Comments and Suggestions for Authors

Spoto et al have a well laid out comprehensive review of adrenomedullin (ADM) expression in patients with viral and bacterial infections.  While the review paper has a ton of relevant information most of the manuscript reads like a point form presentation rather than a written manuscript as most of the paragraphs consist of 1-2 sentences with the overall section lacking both coherence and cohesion.  It needs to be re-written so the text flows better and has better logical flow with cohesive and coherent text.

Major/Overarching Issues

·        Throughout the paper the authors have stated “studies have shown …” but there are no references to those studies.  Every instance referring to a study or that shows data/numbers needs a reference.

·        The references in the text should be on the sentence referring to that study and not only placed at the end of the paragraph.  P7L324 is an example where a pile of references are placed at the end of the paragraph but should be within the paragraph and only selected references used.

·        P3 L102-105 state that ADM is a biomarker and the assay is low cost with a rapid turn around time, but there is no mention/details on the assay used to detect ADM.  There are multiple assays for ADM from ELISA to rapid tests yet this was not discussed and when values of ADM loads were given in the text the assay used was not mention.  Please compare the different tests and how they compare.  Which one is the gold standard and most reliable?

·        Section 1 is written in prose but it lacks logical flow and coherence.  The incidence of influenza and COVID was stated but there is no evidence given for increased ADM in these patients.

·        Section 2 – this is where the writing style changes to 1-2 sentence paragraphs that seem to be like point form from a PowerPoint presentation that makes the writing difficult to follow.

·        Sections 3, 4, 5, 11, and the conclusion are all point form.

·        Figure 1 – This figure is detailed and difficult to follow.  It appears to be in 3 parts with the top red/blue section causing both the left and right flow charts at the bottom.  Perhaps divide into parts A, B, and C.  Is this your hypothesis/model of how things work biologically or was this taken from the literature?  If from the literature please cite your source.  If yours state how you formed the model.  In the blue boxes, right side, second row “bacterial/fungal growth” is difficult to read and has a typo.

·        Figure 2 – another really intricate figure.  I’m puzzled if this figure is for viral or bacterial infections.  You state viral and bacterial infections in the top left but under TLR state “viral RNA” and under Lymphocyte B state “produces spike glycoprotein”.  There are also many abbreviations that are not defined.

·        Most of the caption of Figure 2 is copy/paste on P11L423-444.  This is a serious problem.

·        P12L462-484 needs references

·        P15L580 “MR-proADM stratifies COVID-19 patients”.  What do you mean by stratifies?  Please explain and expand.

·        P15L587-593 This needs to be expanded.  You take the time to explain how ADM works for Influenza and COVID but only use 1-2 sentences for bacteria.  This should be further explained.  How are viral and bacterial infections similar/different?

·        P15L595-603 suggests that ADM is higher in patients with heart involvement, but there’s not much difference between 50% and 53%.  This doesn’t appear to be like much of a difference.

·        P15L595 with respect to COVID what about ADM in patients who are vaccine injured from the COVID vaccines?  Nakahara et al 2023 showed a huge difference in heart activity in unvaccinated and vaccinated patients.  Is the increase in COVID patients simply due to the virus or also from being vaccinated?  Refer to https://pubs.rsna.org/doi/full/10.1148/radiol.230743

·        Section 11, P30 L844-845 This sentence does not make sense and the whole next paragraph lacks logical flow.

·        There is no clear discussion on the future directions where scientific and clinical research needs to focus.

·        Conclusion section – this is not a conclusion paragraph but 5 points.  It needs to be fleshed out in detail.

Minor/Detailed Points to Improve

·        P1L33 – last sentence fragment needs to start with …,  “as well as” antagonizing …

·        P4L168 reference needed for this data.

·        P4L174 “expressed in descending order” what do you mean by this?  It is unclear.

·        P5L216-217 why the underlined text?

·        P9L382 should be SARS-CoV-2 and not COVID-19 as COVID is not a virus.

Comments on the Quality of English Language

The English is decent but needs proofreading as there are several grammatical errors.

Author Response

Spoto et al have a well laid out comprehensive review of adrenomedullin (ADM) expression in patients with viral and bacterial infections.  While the review paper has a ton of relevant information most of the manuscript reads like a point form presentation rather than a written manuscript as most of the paragraphs consist of 1-2 sentences with the overall section lacking both coherence and cohesion.  It needs to be re-written so the text flows better and has better logical flow with cohesive and coherent text.

We thank the Reviewer for in-depth study of our paper. We thank the Reviewer for allowing us to improve the paper which has been almost completely rewritten according to his valuable suggestions. We believe in his suggestions which we appreciated. We hope you can appreciate the improved expository clarity, fluency and greater consistency and harmony of writing.

In particular, the introduction has been completely rewritten.

Suitable references have been inserted into the text.

All missing abbreviations have been inserted in the text, at the foot of the figures and at the end of the review.

We have reconfigured and corrected Figures 1 and 2, renamed 2 and 3 because a new Figure 1 was inserted. Current Figures 2 and 3 have been corrected. All figure legends have been appropriately edited.

We have greatly improved the English form.

We also attach the revised manuscript with the corrections highlighted in yellow and the point by point corrections highlighted in red.

Major/Overarching Issues

  • Throughout the paper the authors have stated “studies have shown …” but there are no references to those studies.  Every instance referring to a study or that shows data/numbers needs a reference.

We thank the Reviewer. Suitable references have been inserted into the text.

  • The references in the text should be on the sentence referring to that study and not only placed at the end of the paragraph.  P7L324 is an example where a pile of references are placed at the end of the paragraph but should be within the paragraph and only selected references used.

We thank the Reviewer. Suitable references have been inserted into the text.

  • P3 L102-105 state that ADM is a biomarker and the assay is low cost with a rapid turn around time, but there is no mention/details on the assay used to detect ADM.  There are multiple assays for ADM from ELISA to rapid tests yet this was not discussed and when values of ADM loads were given in the text the assay used was not mention.  Please compare the different tests and how they compare.  Which one is the gold standard and most reliable?

We thank the Reviewer. We have added cost, turn around time of the only currently available method for biomarker assay.  We have corrected the text appropriately, as follows: “.

In the Introduction: “Therefore, given the high level of diagnostic, prognostic, early-elevation sensitivity, proportionate cost, rapid turn around time [33] of this biomarker, and the incidence and mortality of serious infections such as Influenza [34,35] and SARS-CoV-2 [26] or sepsis [37], it represents a significant clinical aid”.

In the section 3: “MR-proADM analysis is determined by the automated B.R.A.H.M.S. KRYPTOR compact PLUS method (Thermo Fisher Scientific, Hennigsdorf, Germany) using the Time-Resolved Amplified Cryptate Emission (TRACE) technique. The detection limit of the assay is 0.05 nmol/L, and intra- and inter-assay coefficients of variation were under 4 and 11%, respectively [26,33]. The turn around time is about an hour and the cost varies and is about 20 $/test. In healthy individuals, MR-proADM levels are around 0.33 nmol/L [33]”.

  • Section 1 is written in prose but it lacks logical flow and coherence.  The incidence of influenza and COVID was stated but there is no evidence given for increased ADM in these patients.

We thank the Reviewer. We have corrected the text appropriately, as follows: “Indeed, considering that out of about one billion cases of seasonal Influenza per year, at least 3 to 5 million are severe, causing 290.000 to 650. 000 respiratory deaths each year [34,35] and that the more than 772 million confirmed cases of Coronavirus disease (COVID-19) caused more than 6.9 million deaths globally [36] and that ADM has been shown to be the best biomarker for the stratification of mortality risk in critically ill patients with COVID-19 [26], its essential value is appreciated..

  • Section 2 – this is where the writing style changes to 1-2 sentence paragraphs that seem to be like point form from a PowerPoint presentation that makes the writing difficult to follow.
  • Sections 3, 4, 5, 11, and the conclusion are all point form.

We thank the Reviewer for the opportunity that allowed us to greatly improve the manuscript appropriately. We have rewritten the indicated sections and conclusions appropriately. We attach the revised manuscript with the corrections highlighted in yellow and the point by point corrections highlighted in red.

  • Figure 1 – This figure is detailed and difficult to follow.  It appears to be in 3 parts with the top red/blue section causing both the left and right flow charts at the bottom.  Perhaps divide into parts A, B, and C.  Is this your hypothesis/model of how things work biologically or was this taken from the literature?  If from the literature please cite your source.  If yours state how you formed the model.  In the blue boxes, right side, second row “bacterial/fungal growth” is difficult to read and has a typo.

We thank the Reviewer for taking time and attention to study the very important figures for the review. Following his suggestions, we divided Figure 1 into part A, part B and part C, respecting physiology in light blue color and pathology in pink. We hope the improvement can be appreciated. Typographical error has also been corrected.

  • Figure 2 – another really intricate figure.  I’m puzzled if this figure is for viral or bacterial infections.  You state viral and bacterial infections in the top left but under TLR state “viral RNA” and under Lymphocyte B state “produces spike glycoprotein”.  There are also many abbreviations that are not defined.

We thank the Reviewer for allowing us to reflect further in order to improve through graphics the expressed concepts as well. A bacterial TLR (TLR 4) has been included and the actions of B lymphocytes on viral and bacterial infections specified.

  • Most of the caption of Figure 2 is copy/paste on P11L423-444.  This is a serious problem.

All figure legends have been appropriately modified.

  • P12L462-484 needs references. The appropriate reference has been inserted. “ Specifically, ACE2 transforms proinflammatory components of the RAAS axis (angiotensin I and II) into anti-inflammatory components such as Ang 1-9 and especially Ang 1-7 (138)”.
  • P15L580 “MR-proADM stratifies COVID-19 patients”.  What do you mean by stratifies?  Please explain and expand.

We thank the Reviewer for the opportunity to explain this concept further. We modulated as follows: " COVID-19 provides an example of systemic disease that compromises widespread endothelial damage with multiple organ dysfunction syndrome in severe cases and resulting increase in ADM expression [13,163].

MR-proADM, therefore, represents a marker of endothelitis predicting COVID-19 severity [53,164]. In COVID-19 patients, MR-proADM stratifies patients with worse prognosis at risk of major organ damage, including ARDS, with greater need for oxygen therapy, at risk of ICU transfer, in which monoclonal antibody (Adrecizumab) therapy may be efficacy. [13,159,165] "..

  • P15L587-593 This needs to be expanded.  You take the time to explain how ADM works for Influenza and COVID but only use 1-2 sentences for bacteria.  This should be further explained.  How are viral and bacterial infections similar/different?

We thank the Reviewer for noting this discrepancy. We have included additional studies on this issue.

MR-proADM demonstrated a strong capability to detect bacterial pneumonia, particularly in patients with more severe and complex clinical conditions. This finding was further supported by a significant and positive correlation between MR-proADM and severity index score such as pneumonia severity index (PSI) score values. This correlation solely represents the severity of pneumonia [167]. The same conclusions were confirmed by a study of patients with community-acquired pneumonia (CAP) in the emergency department (ED), in which the highest MR-proADM values were found in patients with more severe pneumonia (multilobar rather than unilateral), positive blood culture or who died (2.1 vs. 1.0 nmol/L). The MR-proADM value of 1.8 nmol/L showed a sensitivity of 80% and specificity of 72%, a positive likehood ratio of 2.9 and negative likehood ratio of 0.28 in prognosticating one-year mortality. The researchers conclude that the MR-proADM level on admission predicts the severity and outcome of the CAP patient and can support the role of PSI in therapeutic or hospitalization decision making, showing itself to be a useful tool for risk stratification of the CAP patient [168]. Therefore, MR-proADM succeeds in stratifying patients with both viral and bacterial severe lung infections.

  • P15L595-603 suggests that ADM is higher in patients with heart involvement, but there’s not much difference between 50% and 53%.  This doesn’t appear to be like much of a difference.

We thank the Reviewer.    The meaning of the sentence is that: “Heart involvement (inflammatory, ischemic, thromboembolic, arrhythmic) during viral infections is much more represented than the known prevalence. Postmortem studies and also a recent PET CT study help in improving prevalence information, as noted in the revised text.

Indeed, not all patients with influenza or SARS-COV-2 infection, bacterial pneumonia, or sepsis perform electrocardiograms, echocardiograms, or level III imaging methods.  Specifically, in critically ill patients with viral infections, particularly with SARS-CoV-2, acute cardiac injury is found in up to 15-50% of cases, represented by myocardial injury, endotheliitis, heart failure, Takotsubo cardiomyopathy, acute coronary syndromes, pulmonary thromboembolism, and arrhythmias [166,169–174].

The high rate of acute cardiac injury caused by SARS-CoV-2 is similar to that observed in other viral infections, such as influenza, where myocardial damage was found in 0-53% of cases without any symptoms of cardiac involvement and about 50% of patients had an abnormal electrocardiogram. This damage was also detected during postmortem examinations through the finding of myocarditis, pericarditis, or acute coronary syndrome [166,169–174].

  • P15L595 with respect to COVID what about ADM in patients who are vaccine injured from the COVID vaccines?  Nakahara et al 2023 showed a huge difference in heart activity in unvaccinated and vaccinated patients.  Is the increase in COVID patients simply due to the virus or also from being vaccinated?  Refer to https://pubs.rsna.org/doi/full/10.1148/radiol.230743

We thank the Reviewer for this very interesting food for thought and update.

There are currently no studies that have correlated ADM with SARS-CoV-2 vaccination. It would be interesting to evaluate it.

However, we have included in the text as follows: " In addition, a very interesting retrospective study showed an increase in myocardial 18F-FDG uptake at PET/CT in patients underwent this method of imaging for noninfectious indications other than myocardial inflammation after received the second dose of anti-SARS-CoV-2 vaccine mRNA within 180 days, compared with unvaccinated patients. Unfortunately, a correlation between the increase in myocardial 18F-FDG uptake and the presence of symptoms, clinical or instrumental signs or biohumoral examinations, suggestive of ongoing SARS-CoV-2 myocarditis or other infections, is unknown [181]."

  • Section 11, P30 L844-845 This sentence does not make sense and the whole next paragraph lacks logical flow.

We thank the Reviewer for the valuable interpretation of the text. We have reworded the entire section 11, as follows:

  1. Role of new experimental endothelial barrier stabilization drugs affecting ADM expression

From the evidence reported so far about ADM expression following potentially fatal endothelial barrier damage, we infer the importance of preserving the integrity of the endothelial barrier and the need to develop drugs that can stabilize it despite damage established by the pathogenic noxa.  A recently emerged drug with this goal is a monoclonal antibody (Adrecizumab) that has been used advantageously in small groups of critically ill patients with SARS-CoV-2 infection, sepsis, acute heart failure, or inflammatory bowel disease [41,212]. The critical importance of endothelial integrity directs the formulation of new drugs aimed at preserving vascular barrier integrity during systemic inflammation.

Due to its significant contribution to the pathophysiology of numerous diseases, ADM has been identified as a primary candidate to be targeted with this monoclonal antibody in cases of SARS-CoV-2 infection, sepsis, acute heart failure, and inflammatory bowel disease [42,215].

Adrecizumab, a 160-kDa humanized monoclonal antibody acts by binding to ADM without blocking it and thus therefore without neutralizing its effects, could improve vascular integrity, reduce tissue congestion and consequently improve clinical outcomes [42,215]. It is unable to diffuse freely from the bloodstream to surrounding tissues. When administered, it causes a dose-dependent increas in plasma ADM bound to the antibody. By binding to the N-terminal side of ADM, the monoclonal antibody transfers ADM from the interstitial space to vascular cells, thereby improving vascular integrity and preventing leakage. As a result, this mechanism effectively traps ADM in the bloodstream [41,212]. It also protects against proteolytic degradation of the N-terminal side, thereby increasing the half-life of ADM. Maintained in circulation, ADM protects the function of ECs through binding the ADM receptor formed by the CRLR/RAMP2/3 complex and subsequent stabilization of adhesion junctions and cytoskeleton, while vasodilator properties on VSMCs via the cAMP/PKA pathway in the interstitium may be reduced [216]. Furthermore, in preclinical studies in animals treated with adrecizumab, levels of angiopoietin 1 increased [159]. This peptide is an agonist of EC tight junction integrity and function.

Adrecizumab, a humanized monoclonal antibody that binds to ADM but does not neutralize its effects, has the potential to improve vascular integrity, reduce tissue congestion, and consequently improve clinical results [42,215].

Adrecizumab, a 160 kDa monoclonal antibody, is unable to diffuse freely from the bloodstream to surrounding tissues. When administered, it causes a dose-dependent increas in plasma ADM bound to the antibody. By binding to the N-terminal side of ADM, the non-blocking monoclonal antibody adrecizumab transfers ADM from the interstitial space to the vascular cells, thereby enhancing vascular integrity and preventing leakage. Consequently, this mechanism efficiently traps ADM within the bloodstream [42,215].

It also protects against proteolytic degradation of the N-terminal side, thereby increasing the half-life of ADM. Maintained in circulation, ADM protects the function of ECs through binding the ADM receptor formed by the CRLR/RAMP2/3 complex and subsequent stabilization of adhesion junctions and cytoskeleton, while vasodilator properties on VSMCs via the cAMP/PKA pathway in the interstitium may be reduced [216].

An enlightened early study of eight critically ill patients with COVID-19 and life-threatening ARDS, Adrecizumab administration was followed by a favorable outcome [165,217].

In the phase 2 AdrenOSS-2 trial, in the 300 septic shock patients with elevated ADM (>70 pg/mL) and less than 12 hours after starting treatment with vasopressors for septic shock, who received a single infusion of adrecizumab (2 or 4 mg/kg body weight) or placebo, the latter was safe and well tolerated, and resulted in improved organ function and reduced mortality at day 28 from 28% to 24% [217-219]. In the phase 2 AdrenOSS-2 trial, 300 septic shock patients with elevated ADM (>70 pg/mL) and < 12 h of vasopressor start for septic shock received either a single infusion of adrecizumab (2 or 4 mg/kg b.w.) or placebo, adrecizumab showed itself safe, well tolerated, determined improvement in organ function and reduced mortality at day 28 from 28% to 24% [218–220].

More information will come from a new study in which septic patients are separated from patients with septic shock (a phase 2b/3 study, ENCOURAGE) [1,218].

Further information is needed to determine the ideal time for administration of adrecizumab. During sepsis, the ideal time for administration of this monoclonal antibody should be before the establishment of endothelial barrier damage and thus passage of ADM into the interstitium and could be indicated at the biohumoral level by the significant increase in PCT [1,216,221]. Indeed, as previously reported ADM and PCT belong to the same family as CGRP, which also plays a role in inflammation [46–48].

A new alternative therapeutic strategy to adrecizumab for individualized treatment to protect the integrity of the vascular barrier during sepsis is to inhibit elevated levels of PCT [222,223].  This therapeutic effect could be achieved by two therapeutic approaches. One is through the use of intravenous sitagliptin, which inhibits dipeptidyl peptidase 4 (DPP4) by blocking PCT activation and the establishment of vascular barrier damage. The other is through the administration of olcegepant, which by blocking CRLR/RAMP1 preserves the integrity of the endothelial barrier, in murine sepsis.

Going into more detail, in the plasma of patients with sepsis, PCT is present in two different forms: one is a peptide composed of 116 amino acids and the other is a shortened variant of 114 amino acids [224,225]. Notably, enzymatic cleavage (from 116 to 114 amino acids) is facilitated by DPP4 through the removal of the two amino acids alanine and proline from the N-terminus [154]. This truncated form binds to the CRLR/RAMP1 complex on endothelial cells, causing phosphorylation of VE-cadherin and disruption of its assembly leading to vascular leakage [154,223].

Furthermore, the actions of PCT affect the CGRP receptor, which consists of a heterodimer composed of CRLR and RAMP1 [223,226,227]. PCT has been found to activate the Src pathway leading to phosphorylation of VE-cadherin at tyrosine 685 and separation of the adaptor protein p120 from VE-cadherin in ECs [154].

Recent data have showen that elevated PCT levels after major surgery are also indicative for organ dysfunction, signs of postoperative capillary leakage and increased fluid and vasopressors requirements, and suggest impaired of microvascular integrity [223,228–230]. PCT concentrations of 10 ng/mL, commonly found in individuals with sepsis, trigger the development of intense pulmonary edema in healthy wild type mice [225].

In mice with sepsis, administration of an antibody against the N-terminal side of PCT resulted in reduced lung inflammation and increased survival [222]. Thus, inhibiting procalcitonin's actions by blocking DPP4 with sitagliptin or blocking CRLR/RAMP1 with olcegepant in murine polymicrobial sepsis specifically protects the integrity of the endothelial barrier.

Regarding the ideal timing of sitagliptin administration, in preclinical animal (mouse) studies, ideal intravenous administration of sitagliptin 6 hours after the onset of polymicrobial sepsis disease promises outstanding results [216,221]. Currently, glyptins are only approved for oral administration, but because of the gastroparesis and alterations in oral bioavailability that occur in septic patients, studies have been conducted in healthy volunteers on intravenous administration of sitagliptin, which was well tolerated and showed protective effects on the vessels, heart, and kidneys [6,9,72,231–237].

An alternative innovative strategy for the personalized treatment of widespread inflammation is to focus on inhibiting elevated levels of PCT [222]. In mice with sepsis, the administration of an antibody against the N-terminal side of PCT resulted in reduced inflammation in the lungs and increased survival [222].

Further useful information to determine the ideal timing of adrecizumab administration could provide a promising new use of intravenous sitagliptin, which by inhibiting Dipeptidyl peptidase 4 (DPP4) goes on to block PCT activation allowing vascular barrier control.

Administration of adrecizumab prior to the establishment of endothelial barrier damage and thus passage of ADM into the interstitium could be indicated at the biohumoral level by the increased PCT expression of septic infection and/or also endothelial damage [1,216,221].

In preclinical animal (mice) studies, ideal administration of sitagliptin intravenously 6 hours after disease onset of polymicrobial sepsis would promise outstanding results [216,221].

Recent data have showen that elevated PCT levels after major surgery are also indicative for organ dysfunction, signs of postoperative capillary leakage and increased fluid and vasopressors requirements, and suggest impaired of microvascular integrity [223,228–230]. PCT concentrations of 10 ng/mL, commonly found in individuals with sepsis, trigger the development of intense pulmonary edema in healthy wild type mice [238].

In the plasma of patients with sepsis, PCT is present in two different forms: one is a peptide that consists of 116 amino acids and the other is a shortened variant [224,225]. Enzymatic cleavage is facilitated by DPP4 through the removal of the two amino acids alanine and proline from the N-terminus [154]. The actions of PCT affect the CGRP receptor, which is composed of a heterodimer comprised of CRLR and RAMP1 [223,226,227]. PCT has been found to activate the Src pathway leading to phosphorylation of VE-cadherin at tyrosine 685 and separation of adaptor protein p120 from VE-cadherin in ECs.

Specifically, DPP4 facilitates the truncation of PCT, which results in a bioactive variant containing 114 amino acids. This truncated form binds to the CRLR/RAMP1 complex on endothelial cells, causing phosphorylation of VE-cadherin and disruption of its assembly. This ultimately leads to vascular leakage.

Inhibiting procalcitonin's actions by blocking DPP4 with sitagliptin or blocking CRLR/RAMP1 with olcegepant in murine polymicrobial sepsis specifically protects the integrity of the endothelial barrier.

Currently, glyptins are only approved for oral administration, but because of the gastroparesis and alterations in oral bioavailability that occur in septic patients, studies have been conducted in healthy volunteers on intravenous administration of sitagliptin, which was well tolerated and showed protective effects on the vessels, heart and kidneys [9,12,72,231–237].

Furthermore, some preclinical studies and still few clinical trials are set on the interference of angiopoietin-Tie2 signaling axis, ADM or vascular endothelial (VE-) cadherin precisely to limit endothelial hyperpermeability and its secondary complications [1].  Monoclonal antibody therapy ABTAA clusters the Tie2 receptor, bringing the axis from Tie2 antagonist to Tie2 agonist with benefits on endothelial barrier and endothelial glycocalyx integrity and survival [239]. Instead, administration of ACEI, ARB, or sacubitril/valsartan may play a role in counteracting the increase in RAAS activity brought about by ACE2, but the hypotensive effect and oral formulation do not allow timely administration of these drug classes survival [10, 138].  The experimental administration of Angiotensin 1-7 in an experimental shock model found beneficial effects [141,142,240].

Even considering advanced futuristic treatments, it’s remarkable to note that vaccination remains the most effective strategy for preventing the development of severe viral illnesses in both adults and children. Efficacy rates are achieved of 50–60% with vaccines against Influenza and of 94-95% with mRNA vaccines against SARS-CoV-2, respectively [241–243]. Vaccination is particularly beneficial for high-risk individuals such as those over 65 years of age, young children, those with existing health conditions, and those with compromised immune systems. Furthermore, vaccination could potentially prevent cardiovascular damage, reduce mortality rates, and prevent bacterial coinfections in accordance with antimicrobial stewardship directive [162,193,242,244].

.

  • There is no clear discussion on the future directions where scientific and clinical research needs to focus.

We thank the Reviewer for the suggestions. We have reworded the sentences, as follows:

  • Conclusion section – this is not a conclusion paragraph but 5 points.  It needs to be fleshed out in detail.

We thank the Reviewer for the suggestions. We have reworded the sentences, as follows: "  

"12. Concluding Remarks and Future Perspectives

Regarding the evidence reported so far, we can conclude by asserting that ADM reflects the level of oxidative stress at the endothelial barrier and thus the degree of endotheliitis, the prediction of organ damage, diagnosis and prognosis of severe viral and bacterial infections, and that from the clinical point of view, the integration of clinical signs, clinical scores and value of ADM identifies the phenotype of patients with severe viral and bacterial infections with worse prognosis. The use of such a biomarker thus allows stratification of patients requiring early and intensive management.

We propose the inclusion of MR-proADM as part of the panel of biomarkers needed for the diagnosis and management of critically ill patients. We also hope that experimental studies on endothelial permeability during systemic inflammation can be implemented, which is an essential, fascinating and not fully studied field.

Minor/Detailed Points to Improve

  • P1L33 – last sentence fragment needs to start with …,  “as well as” antagonizing …

We thank the Reviewer. We have reworded the sentences, as follows: "  As well as improving vascular integrity and decreasing vascular permeability, ADM acts as a vasodilator, positive inotrope, diuretic, natriuretic and bronchodilator, antagonizing angiotensin II by inhibiting aldosterone secretion”.

  • P4L168 reference needed for this data.

Appropriate reference has been included. In healthy individuals, MR-proADM levels are around 0.33 nmol/L [33].

  • P4L174 “expressed in descending order” what do you mean by this?  It is unclear. We thank the Reviewer. We have reworded the sentences, as follows: "

Where is ADM synthesized? ADM is widely synthesized throughout the body, but it is particularly abundant in certain organs where it is expressed according to a descending gradient from, such as the adrenal medulla, cardiac atria, lung, kidney, blood vessels, bone, adipose tissue, anterior pituitary, thalamus and hypothalamus [55,56]. ADM is synthesized by different cell types, comprising ECs, VSMCs, macrophages, and monocytes after being exposed to inflammatory triggers like interleukin-1 (Il-1), tumor necrosis factor (TNF), or lipopolysaccharide (LPS) [1,25,44,57,58].

  • P5L216-217 why the underlined text? We removed underlining and improved text comprehension, as follows:

“ Belowe we examine in detail the roles that ADM plays. Its main role is as a vasodilator (Figure 1). Physiologically, as ADM is present both within ECs (intravascular) and in its interstitium and VSMCs, it plays this vasodilator role both intravascularly and interstitially.  In vitro, at the intravascular level it acts on ECs by improving vascular integrity and reducing vascular permeability [5]. At the intravascular level, ADM causes vessel release by stimulating the production of endothelial nitric oxide synthase (eNOS), resulting in vasodilation of VSMCs [1,67-69]. In addition, ADM acts directly on VSMCs by activating protein kinase A (PKA) through the cAMP pathway, resulting in relaxation of VSMCs [1,67-69]. It also maintains vascular integrity by acting on endothelial cell junctions, decreasing actomyosin contraction and vascular permeability during severe inflammation [4,6,10,70].

At the interstitial level, on the other hand, ADM results in vasodilation through (a) inhibition of VSMC contraction, acting both directly on VSMCs and indirectly on ECs and (b) nitric oxide (NO) production [10,71].

  • P9L382 should be SARS-CoV-2 and not COVID-19 as COVID is not a virus.

We thank the Reviewer. We have reworded the sentences, as follows: "  As for the mechanism by which viral and bacterial pathogens cause damage to the intravascular compartment, much has yet to be defined. It appears that both of these groups of pathogens can adhere to the cell surface through the angiotensin-converting enzyme2 receptor (ACE2r), which causes their entry into the cell [136].

It will be considered how viral and/or bacterial infection cause damage in the intravascular compartment (Figure 2) and direct injury to ECs of myocardiocytes and perimycytes (Figure 3). Specifically, in case of SARS-CoV-2 infection (used as the most studied example of viral infection), the virus adheres to the cell surface through the angiotensin-converting enzyme 2 (ACE2)-receptor, which causes it to enter the cell [136].

The English is decent but needs proofreading as there are several grammatical errors.

We thank the Reviewer. We have greatly improved the English form.

Reviewer 2 Report

Comments and Suggestions for Authors

Dear Authors,

Thank you for your well-written manuscript, presenting an interesting issue which is the role and action of adrenomedullin in endothelitis and organ damage in severe viral and bacterial infections. Please pay attention to the following questions, pertaining to your manuscript:

1.      Paragraph 4: Please provide a graphical overview of the mechanisms, triggered by adrenomedullin, at the intravascular and interstitial level.

2.      Figure 1. Please correct the word: ipoxia (hypoxia)

3.      Please check that all the abbreviations are fully explained by their first appearance in the text.

4.      Line 699. Please correct as such: correlating weakly with lactate.

5.      Is there any research about the influence of extracorporeal organ supporting systems (CRRT, ECMO, Cytosorb filter) on the levels of adrenomedullin in septic patients? Please consider to add this information as a paragraph in your text.

Best Regards

Comments on the Quality of English Language

Minor changes

Author Response

Dear Authors,

Thank you for your well-written manuscript, presenting an interesting issue which is the role and action of adrenomedullin in endothelitis and organ damage in severe viral and bacterial infections. Please pay attention to the following questions, pertaining to your manuscript:

  1. Paragraph 4: Please provide a graphical overview of the mechanisms, triggered by adrenomedullin, at the intravascular and interstitial level.

We thank the Reviewer for the insight into our work and his words of appreciation.

A new Figure 1 was inserted to explain the pathophysiological mechanisms of the vasodilatory effect of ADM at the intravascular and interstitial levels. Therefore, we have reconfigured and corrected Figures 1 and 2, renamed 2 and 3 because. Current Figures 2 and 3 have been corrected. All figure legends have been appropriately edited.

We have greatly improved the English form.

  1. Figure 1. Please correct the word: ipoxia (hypoxia) We thank the Reviewer. We have corrected the mistake.
  2. Please check that all the abbreviations are fully explained by their first appearance in the text.

We thank the Reviewer. All missing abbreviations have been inserted in the text, at the foot of the figures and at the end of the review.

  1. Line 699. Please correct as such: correlating weakly with lactate. We thank the Reviewer. We reworded as follows:

MR-proADM would not be shown to be an indicator of anaerobic metabolism by correlating weakly with lactates [185].

  1. Is there any research about the influence of extracorporeal organ supporting systems (CRRT, ECMO, Cytosorb filter) on the levels of adrenomedullin in septic patients? Please consider to add this information as a paragraph in your text.

We thank the Reviewer for this important food for thought. Critically ill patients may require extracorporeal organ supporting systems (CRRT, ECMO or Cytosorb filter). Such organ support systems have shown benefit on the removal of cytokines and thus the inflammatory cascade.  Currently, no studies are available that have evaluated the behavior of ADM in critically ill or septic patients who have received treatment with one of the extracorporeal organ supporting systems such as CRRT, ECMO or Cytosorb filter. Thus, no firm evidence is available on how these systems may interfere with the removal or reduction of ADM values. On the other hand, since ADM is not a cytokine whose increase results in harmful effects, but rather a telltale hormone of endothelial damage that is teleologically elevated for protective purposes, its removal by such systems would not be desirable. For this reason we thought we would supersede the inclusion of a specific paragraph on this subject so as not to be confusing.

However, we reported the relevant literature regarding the Influence of extracorporeal organ support systems (CRRT, ECMO, Cytosorb filter) on adrenomedullin levels in septic patients or COVID-19 patients.

We also attach the revised manuscript with the corrections highlighted in yellow and the point by point corrections highlighted in red.

Mehta Y, Paul R, Ansari AS, Banerjee T, Gunaydin S, Nassiri AA, Pappalardo F, Premužić V, Sathe P, Singh V, Vela ER. Extracorporeal blood purification strategies in sepsis and septic shock: An insight into recent advances. World J Crit Care Med. 2023 Mar 9;12(2):71-88. doi: 10.5492/wjccm.v12.i2.71. PMID: 37034019; PMCID: PMC10075046.

Patients with veno-venous ECMO support:

Montrucchio G, Sales G, Balzani E, Lombardo D, Giaccone A, Cantù G, D'Antonio G, Rumbolo F, Corcione S, Simonetti U, Bonetto C, Zanierato M, Fanelli V, Filippini C, Mengozzi G, Brazzi L. Effectiveness of mid-regional pro-adrenomedullin, compared to other biomarkers (including lymphocyte subpopulations and immunoglobulins), as a prognostic biomarker in COVID-19 critically ill patients: New evidence from a 15-month observational prospective study. Front Med (Lausanne). 2023 Mar 24;10:1122367. doi: 10.3389/fmed.2023.1122367. PMID: 37035317; PMCID: PMC10080079.

A total of 47 patients (22.5%) undergoing veno-venous ECMO (vv-ECMO) support were enrolled (40 before ICU admission and 7 during ICU stay). Of them, 39 died, resulting in overall mortality of 83% in ECMO patients and statistically higher in comparison to the non-ECMO cohort (p = 0.0003).  MR-proADM trend analysis in the subgroup of patients undergoing vv-ECMO failed to evidence statistically significant differences between surviving and non-surviving patients (p = 0.562), and MR-proADM values on days 3 and 7 suggest a difference between groups, although not statistically significant (p = 0.08). Considering other standard biomarkers, only the lymphocyte count was found significantly different between survivors and non-survivors (p = 0.0471) (Figure 5).

Cytosorb filter:

 Blood Purif 2023;52:183–192 Effect of Hemadsorption Therapy in Critically Ill Patients with COVID-19 (CYTOCOV-19): A Prospective Randomized Controlled Pilot Trial Dominik Jarczak Kevin Roedl Marlene Fischer Geraldine de Heer Christoph Burdelski Daniel Peter Frings Barbara Sensen Olaf Boenisch Pischtaz Adel Tariparast Stefan Kluge Axel Nierhaus

242 Patients with COVID-19, refractory shock (norepinephrine ≥0.2 µg/kg/min to maintain a mean arterial pressure ≥65 mm Hg), interleukin-6 (IL-6) ≥500 ng/L, and an indication for renal replacement therapy or extracorporeal membrane oxygenation were included. Patients received either hemadsorption therapy (HT) or standard medical therapy (SMT). For HT, a CytoSorb® adsorber was used for up to 5 days and was replaced every 18–24 h. The primary endpoint was sustained hemodynamic improvement (norepinephrine ≤0.05 µg/kg/ min ≥24 h). Results: Of 242 screened patients, 24 were randomized and assigned to either HT (N = 12) or SMT (N = 12). Both groups had similar severity as assessed by SAPS II (median 75 points HT group vs. 79 SMT group, p = 0.590) and SOFA (17 vs. 16, p = 0.551). Median IL-6 levels were 2,269 (IQR 948–3,679) and 3,747 (1,301–5,415) ng/L in the HT and SMT groups at baseline, respectively (p = 0.378), at baseline MR-proADM 6.25 nmol/L (4.03–7.26) 10.01 (4.74–12.24) HT group vs. 79 SMT group, p =0.089. Shock resolution (primary endpoint) was reached in 33% (4/12) versus 17% (2/12) in the HT and SMT groups, respectively (p = 0.640). Twenty-eight-day mortality was 58% (7/12) in the HT compared to 67% (8/12) in the SMT group (p = 1.0). During the treatment period of 5 days, 6/12 (50%) of the SMT patients died, in contrast to 1/12 (8%) in the HT group. Conclusion: HT was associated with a non-significant trend toward clinical improvement within the intervention period. In selected patients, HT might be an option for stabilization before transfer and further therapeutic decisions

The primary endpoint of shock reversal within 10 days of randomization was reached by 4 patients (33%) in the HT group and 2 patients (17%) in the SMT group (p = 0.640). The time to shock reversal was 6.3 (3.7–10.0) days in the HT and 9.2 (5.1–15.9) days (p = 0.110) in the SMT group. We observed a 28-day mortality of 58% (n = 7) in the HT group and of 67% (n = 8) in the SMT group (p = 0.382, cf. Kaplan-Meier survival estimates (Fig. 2). Primary and secondary endpoints are shown in detail in Table 3.

Adm assay after treatment was not in the secondary outcomes. IL-6, PCT and D-Dimers were assayed, which were not shown to be significant in the descent.

Pro-adrenomedullin as a clinical predictor after cardiac surgery J Van Fessem, F De Graaf, J Van Paassen, S Arbous LUMC, Leiden, the Netherlands Critical Care 2013, 17(Suppl 2):P32 (doi: 10.1186/cc11970)

Critical Care 2013, Volume 17 Suppl 2 http://ccforum.com/supplements/17/S2

Mehta Y, Paul R, Ansari AS, Banerjee T, Gunaydin S, Nassiri AA, Pappalardo F, Premužić V, Sathe P, Singh V, Vela ER. Extracorporeal blood purification strategies in sepsis and septic shock: An insight into recent advancements. World J Crit Care Med. 2023 Mar 9;12(2):71-88. doi: 10.5492/wjccm.v12.i2.71. PMID: 37034019; PMCID: PMC10075046.

Sepsis is one of the leading causes of mortality in critically ill patients globally. Substantial progress is made in the field of extracorporeal therapies and sepsis. CytoSorb® is emerging as an adjunct therapy to improve hemodynamic stability. This device is an International Organization for Standardization certified, European Conformité Européenne mark-approved class IIb medical device that is designed to remove excess inflammatory cytokines from the blood. There are extensive published reports of its use in the field of septic shock with improved survival rates and other improved biochemical parameters. However, clinical acceptance is still limited due to a lack of large random clinical trials.

Akil A, Ziegeler S, Reichelt J, Rehers S, Abdalla O, Semik M, Fischer S. Combined Use of CytoSorb and ECMO in Patients with Severe Pneumogenic Sepsis. Thorac Cardiovasc Surg. 2021 Apr;69(3):246-251. doi: 10.1055/s-0040-1708479. Epub 2020 Apr 6. PMID: 32252114.

Background High morbidity and mortality are frequently reported in intensive care patients suffering from severe sepsis with systemic inflammation. With the development of severe respiratory failure, extracorporeal membrane oxygenation (ECMO) is often required. In this study, cytokine adsorption therapy in combination with ECMO is applied in patients with acute respiratory distress syndrome (ARDS) due to severe pneumogenic sepsis. The efficacy of this therapy is evaluated compared with a historical cohort without hemoadsorption therapy.

Methods Between January and May 2018, combined high-flow venovenous ECMO and CytoSorb therapy (CytoSorb filter connected to ECMO circuit) was applied in patients (n = 13) with pneumogenic sepsis and ARDS. These patients were prospectively included (CytoSorb group). Data from patients (n = 7) with pneumogenic sepsis and ECMO therapy were retrospectively analyzed (control group).

Results All patients survived in the CytoSorb group, where the 30-day mortality rate reached 57% in the control group. After CytoSorb therapy, we instantly observed a significant reduction in procalcitonin (PCT) and C-reactive protein (CRP) levels compared with the control group. Within 48 hours, the initial high doses of catecholamine could be weaned off only in the CytoSorb group.

Conclusions Our results indicate that CytoSorb in combination with ECMO is an effective therapy to prevent escalation of sepsis with rapid weaning off high-dose catecholamine infusions and quick reduction in PCT and CRP levels. Optimal timing of immunomodulatory therapy and impact on ECMO-related inflammation still need to be furtherly investigated.

Jarczak, D.; Kluge, S.; Nierhaus, A. Septic Hyperinflammation—Is There a Role for Extracorporeal Blood Purification Techniques? Int. J. Mol. Sci. 202425, 3120. https://doi.org/10.3390/ijms25063120

Brouwer, W.P., Duran, S., Kuijper, M. et al. Hemoadsorption with CytoSorb shows a decreased observed versus expected 28-day all-cause mortality in ICU patients with septic shock: a propensity-score-weighted retrospective study. Crit Care 23, 317 (2019). https://doi.org/10.1186/s13054-019-2588-1Background and aims: Innovative treatment modalities have not yet shown a clinical benefit in patients with septic shock. To reduce severe cytokinaemia, CytoSorb as an add-on to continuous renal replacement therapy (CRRT) showed promising results in case reports. However, there are no clinical trials investigating outcomes. Methods: In this investigator-initiated retrospective study, patients with septic shock were treated with CRRT + CytoSorb (n = 67) or CRRT alone (n = 49). The primary outcome was the 28-day all-cause mortality rate. Patients were weighted by stabilized inverse probability of treatment weights (sIPTW) to overcome differences in baseline characteristics. Results: At the start of therapy, CytoSorb-treated patients had higher lactate levels (p < 0.001), lower mean arterial pressure (p = 0.007) and higher levels of noradrenaline (p < 0.001) compared to the CRRT group. For CytoSorb, the mean predicted mortality rate based on a SOFA of 13.8 (n = 67) was 75% (95%CI 71–79%), while the actual 28-day mortality rate was 48% (mean difference − 27%, 95%CI − 38 to − 15%, p < 0.001). For CRRT, based on a SOFA of 12.8 (n = 49), the mean predicted versus observed mortality was 68% versus 51% (mean difference − 16.9% [95%CI − 32.6 to − 1.2%, p = 0.035]). By sIPTW analysis, patients treated with CytoSorb had a significantly lower 28-day mortality rate compared to CRRT alone (53% vs. 72%, respectively, p = 0.038). Independent predictors of 28-day mortality in the CytoSorb group were the presence of pneumosepsis (adjusted odds ratio [aOR] 5.47, p = 0.029), higher levels of lactate at the start of CytoSorb (aOR 1.15, p = 0.031) and older age (aOR per 10 years 1.67, p = 0.034). Conclusions: CytoSorb was associated with a decreased observed versus expected 28-day all-cause mortality. By IPTW analysis, intervention with CytoSorb may be associated with a decreased all-cause mortality at 28 days compared to CRRT alone.

Honoré, Patrick M. MD; Matson, James R. MD. Extracorporeal removal for sepsis: Acting at the tissue level—The beginning of a new era for this treatment modality in septic shock *. Critical Care Medicine 32(3):p 896-897, March 2004. | DOI: 10.1097/01.CCM.0000115262.31804.46

Lee, O. P.1; Lee, O. P.2; Kanesan, N.3; Leow, E. H.4; Sultana, R.5; Chor, Y.1; Gan, C.6; Lee, J.4; Lee, J.5. PP281 [Extracorporeal Support » Total plasma exchange]: THERAPEUTIC PLASMA EXCHANGE IN CHILDREN WITH SEVERE SEPSIS AND SEPTIC SHOCK: A SYSTEMATIC REVIEW AND META-ANALYSIS. Pediatric Critical Care Medicine 23(Supplement 1 11S):, November 2022. | DOI: 10.1097/01.pcc.0000900932.70053.23

P350 Enhancement of usual emergency department care with proadrenomedullin to improve outcome prediction - Results from the multi-national, prospective, observational TRIAGE study

A. Kutz1, P. Hausfater2, D. Amin3, T. Struja1, S. Haubitz1, A. Huber4, B. Mueller1, P. Schuetz1

1Kantonsspital Aarau, Aarau, Switzerland; 2Emergency Department, Groupe Hospitalier Pitié-Salpêtrière, Paris, France; 3Morton Plant Hospital, Clearwater, USA; 4Department of Laboratory Medicine, Kantonsspital Aarau, Aarau, Switzerland

Introduction

Previous studies have found Proadrenomedullin (ProADM), an inflammatory blood marker, to provide additional prognostic information for risk stratification. We aimed to translate ProADM cut-off levels into an easy-to-use emergency department (ED) triage algorithm to improve current risk assessments and clinical outcome prediction of medical patients.

Methods

In this large multi-national, prospective, observational study from Switzerland, France and the United States [1], we combined two ProADM cut-off values with a five-level ED risk assessment tool (Manchester Triage System, MTS) to further risk-stratify medical ED patients [2]. The performance of this algorithm was tested to predict adverse clinical outcome.

Results

Using data from 7132 consecutive ED patients (median age 62 years, 53.3 % male), the risk for 30 day mortality showed a stepwise increase from low (0.64 % [95 % CI 0.37-0.92]) in patients with a ProADM value of <0.75 nmol/L, to intermediate (4.38 % [95 % CI 3.58-5.18]) with a ProADM value >0.75 nmol/L and <1.5 nmol/L, to high (15.5 % [95 % CI 13.5-17.5]) with a ProADM value of >1.5 nmol/L (ANOVA, p < 0.0001). Combining initial ED triage (MTS) and ProADM reveals an observed risk for 30 day mortality of 0.5 % (14/2780 patients) in case of low priority initial triage and a ProADM value of <0.75 nmol/L until 23.1 % (96/416) in case of high priority initial triage and a ProADM value of >1.5 nmol/L. In low priority patients the overall 30 day mortality was 2.95 % (164/5550). Adding ProADM values of <0.75 nmol/L the relative risk reduction was -83 %, whereas adding ProADM values of >1.5 nmol/L increased the risk for mortality four times. In high priority patients (11.56 % overall 30 day mortality) relative risk reduction was -86 % adding ProADM values of <0.75 nmol/L, and doubled in patients with ProADM values of >1.5 nmol/L.

Conclusions

The proposed ProADM based triage algorithm allows a more accurate prediction for adverse clinical outcome in medical ED patients at high risk. To proof safety and efficacy of this new ED triage algorithm, an intervention study involving a rapid ProADM point-of-care measurement is mandatory.

References

  1. Schuetz, P., et al., Crit Care, 2015. 19(1): p. 377.
  2. Steiner, D., et al., J Emerg Med, 2015. (in press)

P019 Relation between adrenomedullin and short-term outcome in ICU patients: Results from the frog ICU study

E. G. Gayat1, J. Struck2, A. Cariou3, N. Deye1, B. Guidet4, S. Jabert5, J. Launay1, M. Legrand6, M. Léone7, M. Resche-Rigon6, E. Vicaut1, A. Vieillard-Baron8, A. Mebazaa1

Introduction: Adrenomedullin (ADM) is a peptide with 52 amino acids which has strong vasodilator activity. Elevated plasma ADM levels have been detected in a wide variety of physiological and pathological conditions, with the highest elevations observed in septic shock. Marino et al. demonstrated a strong association of admission ADM levels with the severity of sepsis, supporting an earlier report describing elevated ADM levels in those with severe sepsis and those with septic shock [1]. By using a novel assay specific for bioactive ADM (bio-ADM) as Marino et al.the aim of the study was to assess the relation between ADM measured at admission and in-ICU mortality in consecutive patients regardless the cause of admission.

Methods: The French and euRopean Outcome reGistry in Intensive Care Units (FROG-ICU) study was a multicenter observational study, including 2087 consecutive patients followed up to one year for those who survived to ICU stay. The protocol has previously been described [2]. Plasma were collected at admission for all patients and at discharge for in-ICU survivors. ADM was measured in all plasma using a sandwich assay specific for bio-ADM. The association between in-ICU mortality and the level of ADM was assessed by univariate analysis and adjusted analysis for severity at admission measured by the SAPS-II. Improvement in area under the ROC curve and reclassification indices were assessed.

Results: 2087 patients have been included, 65 % male with a median age of 63 (51-74), a median Charlson score of 3 (1-5) and a median SAPS-II 49 (36-63). Septic shock was present in 488 (23 %) patients. Median (and interquartile range) of ADM in-ICU survivors and non-survivors was 57 pg/mL [30-114] and 110 pg/mL [63-220], respectively (p < 0.001). Hazard ratio of in-ICU death for patients with a level of ADM higher than the median value was 2.12 (95%CI: 1.73-2.60) and 1.68 (95%CI: 1.36-2.07) when adjusted for SAPS-II. Area under the ROC curve of SAPS-II was significantly improved by the addition of ADM (0.653 [0.624 - 0.682] to 0.702 [0.675 - 0.729], p = 0.01).

Conclusions: In the present study, ADM was independently associated with in-ICU mortality and improved prognostic prediction. The clinical and therapeutic implication of these findings need to be further investigated.References[1] R Marino et al. Critical Care 2014, 18:R34[2] Mebazaa et al. BMC Anesthesiology (2015) 15:143

Reviewer 3 Report

Comments and Suggestions for Authors

Very well-written and structured manuscript. Excellent figures.  It provides all the necessary information on the topic discussed including numerous references.

1        The manuscript is a review article so there is no research question, it summarizes all current knowledge on the pathophysiology of adrenomedullin in viral and bacterial infections.

2        As it is a review article there is no original, unpublished work included. All information provided is from other studies. There is no specific gap addressed but some interesting future perspectives.

3        An excellent, very comprehensive manuscript that includes all current knowledge on the topic discussed, excellent figures.

4        The structure and the methodology of the writing need no improvement, is in a logical manner, reader-friendly.

5        This is not an original work, no experiments are included, the message is very clear regarding the pathophysiological pathways, the involvement of adrenomedullin in the development of endothelitis in severe viral and bacterial infections.

6        The bibliography is very extensive, including old and many new studies related to the topic, no inappropriate self-citations.

7        No tables are included, all 3 figures are very good, and they depict clearly the complexity of the pathophysiological pathways regarding the correlation of ADM expression and endothelitis in both types of infections in a friendly-reader format. The quality of data used is of high level.

Comments on the Quality of English Language

Minor English editing needed e.g line 644 bath both, line 941 seriuous. 

Figure 1 IPOXIA (hypoxia)

Author Response

Very well-written and structured manuscript. Excellent figures.  It provides all the necessary information on the topic discussed including numerous references.

1        The manuscript is a review article so there is no research question, it summarizes all current knowledge on the pathophysiology of adrenomedullin in viral and bacterial infections.

2        As it is a review article there is no original, unpublished work included. All information provided is from other studies. There is no specific gap addressed but some interesting future perspectives.

3        An excellent, very comprehensive manuscript that includes all current knowledge on the topic discussed, excellent figures.

4        The structure and the methodology of the writing need no improvement, is in a logical manner, reader-friendly.

5        This is not an original work, no experiments are included, the message is very clear regarding the pathophysiological pathways, the involvement of adrenomedullin in the development of endothelitis in severe viral and bacterial infections.

6        The bibliography is very extensive, including old and many new studies related to the topic, no inappropriate self-citations.

7        No tables are included, all 3 figures are very good, and they depict clearly the complexity of the pathophysiological pathways regarding the correlation of ADM expression and endothelitis in both types of infections in a friendly-reader format. The quality of data used is of high level.

We thank the Reviewer for appreciating our work and for the valuable suggestions recommended to us. We hope you can appreciate the improved expository clarity, fluency and greater consistency and harmony of writing.

We have greatly improved the English form.

We also attach the revised manuscript with the corrections highlighted in yellow and the point by point corrections highlighted in red.

Minor English editing needed e.g line 644 bath both,

We thank the Reviewer. We have corrected the text appropriately, as follows: “The definition of sepsis-induced myocardial dysfunction indicates left ventricular (LV) systolic dysfunction, whereas both ventricles can be impaired and when right ventricular (RV) systolic dysfunction is present, the prognosis of septic patients is poor [195-198].

line 941 seriuous. We thank the Reviewer. We have corrected the text appropriately, as follows: “Regarding the evidence reported so far, we can conclude by asserting that ADM reflects the level of oxidative stress at the endothelial barrier and thus the degree of endotheliitis, the prediction of organ damage, diagnosis and prognosis of severe viral and bacterial infections, and that from the clinical point of view, the integration of clinical signs, clinical scores and value of ADM identifies the phenotype of patients with severe viral and bacterial infections with worse prognosis. The use of such a biomarker thus allows stratification of patients requiring early and intensive management.

We propose the inclusion of MR-proADM as part of the panel of biomarkers needed for the diagnosis and management of critically ill patients. We also hope that experimental studies on endothelial permeability during systemic inflammation can be implemented, which is an essential, fascinating and not fully studied field.

Figure 1 IPOXIA (hypoxia) We thank the Reviewer. We have reconfigured and corrected Figures 1 and 2, renamed 2 and 3 because a new Figure 1 was inserted. Current Figures 2 and 3 have been corrected. All figure legends have been appropriately edited.
